# THE DELEUZIAN REPRESENTATION HYPOTHESIS

**Clément Cornet, Romaric Besançon & Hervé Le Borgne**
Université Paris-Saclay, CEA, List,
F-91120, Palaiseau, France
{clement.cornet,romaric.besancon,herve.le-borgne}@cea.fr

## ABSTRACT

We propose an alternative to sparse autoencoders (SAEs) as a simple and effective unsupervised method for extracting interpretable concepts from neural networks. The core idea is to cluster differences in activations, which we formally justify within a discriminant analysis framework. To enhance the diversity of extracted concepts, we refine the approach by weighting the clustering using the skewness of activations. The method aligns with Deleuze's modern view of concepts as differences. We evaluate the approach across five models and three modalities (vision, language, and audio), measuring concept quality, diversity, and consistency. Our results show that the proposed method achieves concept quality surpassing prior unsupervised SAE variants while approaching supervised baselines, and that the extracted concepts enable steering of a model's inner representations, demonstrating their causal influence on downstream behavior.

## 1 INTRODUCTION

Interpretability of neural network representations is essential for building trustworthy models, enabling a deeper understanding of the mechanisms underlying a model's predictions, and promoting fairness and accountability. However, interpreting the internal representations learned by neural networks remains a central challenge in deep learning. Sparse autoencoders (SAEs) (Bricken et al., 2023; Cunningham et al., 2023) have emerged as a powerful tool for extracting sparse and semantically meaningful features from model activations. Nevertheless, they face challenges that limit their applicability. Notably, they suffer from difficulties in training, and may still yield polysemantic features, not corresponding to a single interpretable concept. Moreover, sparse autoencoders (and similar methods) rely on feature sparsity as a proxy for interpretability, a choice that has been criticized as potentially inadequate (Sharkey et al., 2025).

We introduce an alternative to sparse autoencoders (SAEs) for extracting features that correspond to interpretable concepts from neural networks. Drawing inspiration from Deleuze's philosophical view of concepts as differences, we model concepts as directions that capture distinctions between representations of individual samples. Specifically, our approach can be seen as an unsupervised

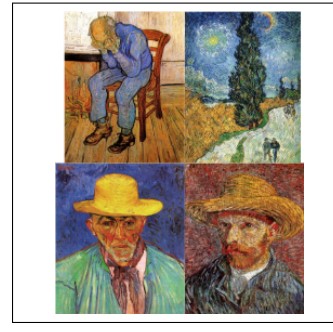

| (a) Image: Van Gogh's Paintings | (b) Text: Sports Achievements | (c) Audio: Brass Instruments |

"Winning the prize"

"the Gold Medal"

"the World Record"

Figure 1: Our method extracts diverse concepts from image, text and audio models.

discriminant analysis: it identifies directions in the internal representation that best separate data samples. We estimate those directions by sampling activation differences between pairs of data points, then use KMeans clustering to uncover recurring patterns. Our analysis is further refined using distributional skewness to promote diversity.

Evaluating interpretability methods remains a major challenge. SAEs are often assessed by their reconstruction–sparsity trade-off, which does not necessarily reflect interpretability. Hence, most recent studies in this field are also evaluated qualitatively, showing their relevance through selected examples. While insightful, such evaluations provide limited support. In contrast, we adopt a quantitative evaluation based on probe loss (Gao et al., 2025), which measures the extent to which extracted concepts capture the attributes expected to be present in a dataset. To ensure robust evaluation, we apply this metric to a broad set of 874 attributes spanning different tasks, five datasets and five models across three modalities (image, text and audio). Our method captures the desired attributes more effectively than recent SAE-based approaches. In several settings, it is competitive with supervised linear discriminant analysis. Beyond the presence of expected attributes, we also evaluate cross-run consistency with the Maximum Pairwise Pearson Correlation (MPPC) (Wang et al., 2025), establishing a comprehensive evaluation framework for concept evaluation methods. Finally, we demonstrate concept steering on text and image models, showing that manipulating extracted concepts causally influence downstream behavior, without incurring information loss.

Hence, the main contribution of this paper is a novel type of approach of mechanistic interpretability of neural networks. We investigate the fundamental principle underlying our approach and demonstrate that it achieves globally more compelling results than state-of-the-art sparse autoencoder (SAE)–based techniques. Our method is advantageous in its simplicity: it is governed by a single, interpretable hyperparameter. The proposed principle is theoretically grounded in discriminant analysis and clustering, and further relates to Deleuze's philosophical notion of "concepts." Similar to SAE-based approaches, our method is fully unsupervised and therefore does not require manual specification or annotation of the identified concepts.

## 2 METHODS

### 2.1 CRITERIA AND CONCEPTUAL GROUNDING

Our aim is to extract an ontology of "concepts" from a neural network, by analyzing its activations. Before proposing our approach, we first discuss the criteria such concepts should satisfy.

- *Interpretability*: this work aims to extract human-interpretable features, that are then referred to as "concepts".

- *Transparency*: in order to gain interpretable insights into the model, the approach itself should be as simple and transparent as possible, not relying on non-interpretable hyperparameters.

- *Diversity*: the extracted concepts should be semantically diverse, in order to represent a wide variety of data samples, ideas, and semantic levels.

- *Consistency*: the approach should consistently yield similar concepts when run multiple times with different random seeds.

Existing methods in mechanistic interpretability typically extract unsupervised concepts by reconstructing model activations (Bricken et al., 2023; Cunningham et al., 2023). Because they are trained to minimize reconstruction error, such approaches are driven to capture as much variance in the activation space as possible, subject to sparsity constraints. This framing implicitly presents concepts as universal structural components of the model activations, echoing the classical philosophical view of concepts as "the universal essence of a fact" (Plato, c. 375 BCE; Hegel, 1816). However, such a representation has been criticized as overly restrictive (Nietzsche, 1889; Sartre & Elkaïm-Sartre, 1946). More recent perspectives instead emphasize concepts as arising from *Difference and Repetition* (Deleuze, 1968), rather than universals. Following this idea, our approach does not attempt to model the full variance of activations. Instead, it identifies recurring differences between activations.

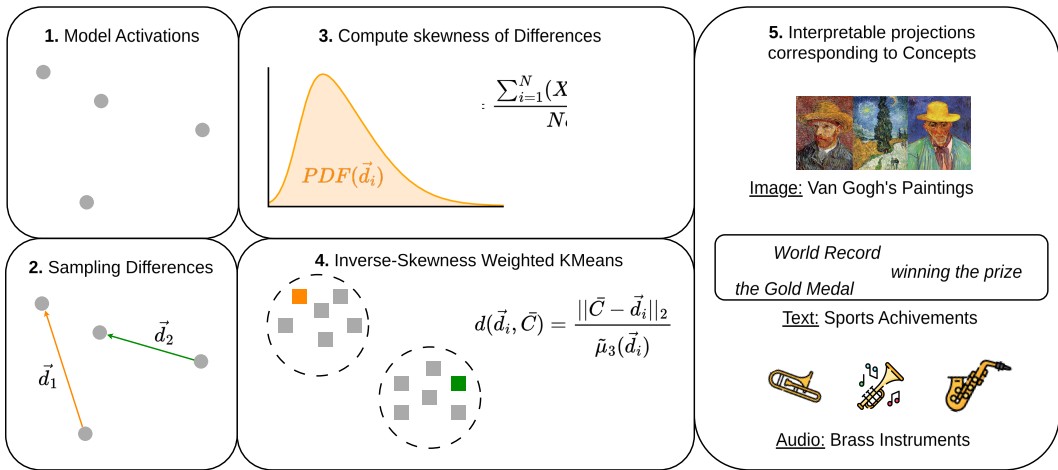

Figure 2: Overview of our concept extraction approach. We sample pairwise differences in activation between samples. Then, we use the inverse-skewness of those differences to selected the final concepts, corresponding to vectors in the activation space.

## 2.2 EXTRACTING REPEATED DIFFERENCES IN ACTIVATION SPACE

Our objective is to extract concepts from model activations, at a given layer with $\mathcal{D}$ dimensions, over a dataset of $N$ samples. To represent repeated differences in activations between data samples, we define $D = \{\vec{d_1}, \vec{d_2}, ..., \vec{d_N}\}$ as a set of $\mathcal{D}$-dimensional pairwise differences in activation between samples. Since our approach is fully unsupervised, we cannot restrain $D$ to contrastive pairs between two classes. However, computing all pairwise differences is quadratic in $N$. To approximate the distribution of differences, we instead randomly sample $N$ pairs, ensuring that each data point is used once on each side of the subtraction.

To constrain our concept dictionary to a fixed number of concepts $k$, we cluster activation differences using KMeans (Lloyd, 1982; Zeng & Zheng, 2019). However, some activation differences exhibit highly skewed distributions: they remain near-zero for most samples, but occasionally spike to large values. Those differences tend to dominate the Euclidean distance used by standard KMeans, and produce redundant clusters (Milligan, 1980). The skewness of a distribution $X$ is its normalized third central moment. For a concept direction $\vec{d_i}$, we consider skewness as that of the projection $\{\vec{d_i} \cdot \vec{x_j}\}_{j=1}^N$

$$\tilde{\mu}_3(\vec{d_i}) = \frac{\sum_{j=1}^N \left( \vec{d_i} \cdot \vec{x_j} - \mu(\vec{d_i} \cdot \vec{x_j}) \right)^3}{N\sigma(\vec{d_i} \cdot \vec{x_j})^3} \tag{1}$$

where $\mu()$. and $\sigma(.)$ are the mean and standard deviation over the $\{x_j\}_{j=1}^N$. Since highly skewed coordinates tend to produce redundant clusters, we penalize them by assigning weights inversely proportional to skewness. In order to avoid ill-defined clustering with negative weights, and to consider opposite directions $\vec{d_i}$ as similar (as we are seeking directions, regardless of their orientation), we consider $-\vec{d_i}$ for differences with negative skewness. This results in a variant of Feature-Weighted KMeans (Huang et al., 2005), in which concept directions are weighted during centroids computation, in order to promote concept diversity. More precisely, this clustering defines the weighted distance between $\vec{d_i}$ and its corresponding centroid $\bar{C}$ as

$$d(\vec{d_i}, \bar{C}) = \frac{1}{\tilde{\mu}_3(\vec{d_i})} ||\bar{C} - \vec{d_i}||_2$$

The obtained centroids are then used as concept vectors.

Both pair sampling and KMeans clustering run in linear time and memory with respect to dataset size $N$ and activation dimension $\mathcal{D}$, demonstrating scalability of our approach towards large datasets, or large models.

Finally, this procedure retains a simple and transparent formulation ( Figure 2), that are key properties for interpretability research. Notably, the number of extracted concepts $k$ is the only hyperparameter required for our approach, and is itself interpretable.

## 2.3 CONNECTION TO DISCRIMINANT ANALYSIS

We aim to extract "concepts" from model activations, defining a concept as a difference between ideas. In a supervised setting, this objective relates closely to discriminant analysis (Fisher, 1936), which identifies a direction $\vec{c}$ orthogonal to the optimal separating hyperplane between two classes. Let $\Sigma_A$ and $\Sigma_B$ be the class covariances, and $\mu_A$ and $\mu_B$ their means. The separation between classes is maximized by:

$$\vec{c} \propto (\Sigma_A + \Sigma_B)^{-1}(\vec{\mu}_A - \vec{\mu}_B) \tag{2}$$

Consider two samples $i$ and $j$ with activations $\vec{x}_i$ and $\vec{x}_j$, and suppose we seek the optimal separation between clusters with means $\vec{x}_i$ and $\vec{x}_j$, distinguished by a concept $\vec{c}$. In high-dimensional spaces (typically $\geq 512$ dimensions for transformers), we approximate $\Sigma_i$ and $\Sigma_j$ as diagonal, containing each dimension's variance (Ahdesmäki & Strimmer, 2010).

From equation 2, $\vec{c} \propto \vec{x}_i - \vec{x}_j$ achieves optimal separation when $\Sigma_i \propto \Sigma_j \propto I$, i.e., under isotropic cluster distributions. Thus, treating activation differences as the optimal separation between ideas is equivalent to assuming isotropic distributions of concepts in activation space.

Unlike standard LDA, equation 2 does not require homoscedasticity or Gaussianity (McLachlan, 2005), and naturally extends to multiclass discrimination (Rao, 1948).

In Appendix G, we derive a quadratic extension to our approach, that accounts for anisotropic distribution of concepts. While theoretically interesting, it does not lead to better experimental results. For this reason, we focus on the isotropic approach (*i.e* $\Sigma_i \propto \Sigma_j \propto I$) in the following.

## 2.4 LOSSLESS STEERING

Sparse autoencoders and related methods allow steering of extracted concepts (Zhou et al., 2025). To do so, they project sample activations in their concept space, apply a steering vector, and projects back into the activation space. The two projections required introduce reconstruction error and information loss. In contrast, our extracted concepts are vectors in the activation space. Therefore, we can perform steering directly in the activations space. To steer the embedding of a sample $x$, with a magnitude $\alpha$ and a concept $\vec{c}_i$, consider its steered representation $\tilde{x} = x + \alpha\vec{c}_i$. If one steers a concept by $+\alpha$, then by $-\alpha$, we retrieve exactly the base activation. By avoiding projections into and out of the concept space, our approach enables lossless steering: the modifications affect only the targeted direction and can be exactly reversed.

## 3 EXPERIMENTS

**Datasets and Models**   To evaluate our concept extraction methods, we conduct a large-scale study spanning five models and five datasets across three modalities (vision, language, and audio), covering a wide variety of semantic attributes.

For text, we use two datasets: IMDB (Maas et al., 2011) and CoNLL-2003 (Tjong Kim Sang & De Meulder, 2003). IMDB provides sentence-level binary sentiment classification labels, while CoNLL-2003 provides token-level labels for named entity recognition (NER), part-of-speech (POS) tagging, and syntactic chunking. For vision, we use a subset of ImageNet (Russakovsky et al., 2015) with 100 classes and the WikiArt dataset (Baylies, 2020) which contains paintings labeled by artist (129 classes), style (27 classes), and genre (11 classes). Concerning text datasets, IMDB has binary classification labels, while CoNLL-2003 has token-wise labels for NER (9 classes), POS-tagging (47 classes) and chunk tags (23 classes). For audio, we use AudioSet (Gemmeke et al., 2017), with multi-classification labels (527 audio classes).

Our text experiments are conducted on DeBERTa (He et al., 2021) and the encoder of BART (Lewis et al., 2020), as well as Pythia-70M (Biderman et al., 2023). For vision, we evaluate DinoV2 (Oquab

et al., 2023) and CLIP (Radford et al., 2021). For audio, we use a pretrained Audio Spectrogram Transformer (AST) (Gong et al., 2021). We only consider encoder models, (including the encoder of BART). This choice allows us to evaluate the quality of extracted concepts with respect to supervised labels that are likely represented at the analyzed layer of each model, since our objective is to compare concept extraction methods. It also enables comparable analyses across multiple modalities. More details on datasets and models are provided in Appendix B.

**Baselines** Sparse autoencoders (SAE) are predominant among concept extraction methods. We compare our method to five different types of SAEs:

- VanillaSAE (**Van-SAE**) (Bricken et al., 2023): standard SAE, trained with an $L_2$ reconstruction loss, and enforcing sparsity via an $L_1$ penalty which requires a coefficient $\lambda$;
- GatedSAE (**Gat-SAE**) (Rajamanoharan et al., 2024a): SAE learning activations gates, hence separating feature selection and magnitude estimation;
- JumpReLUSAE (**JR-SAE**) (Rajamanoharan et al., 2024b): SAE with a learnable threshold $\theta_i$ for each concept, designed to minimize the reconstruction error;
- MatryoshkaSAE (**Mat-SAE**) (Bussmann et al., 2025): SAE learning nested dictionaries of concepts, focusing on hierarchies of concepts, belonging to multiple semantic levels;
- TopKSAE (**Tk-SAE**) (Gao et al., 2025): SAE enforcing sparsity via a *TopK* activation function, that sets every activation to zero, except the $k$ highest.
- ArchetypalSAE (**A-SAE**) (Fel et al., 2025): SAE constraining decoder atoms to be combinations of activations, to gain stability.
- Pretrained Sparse Autoencoders (**Pretrained**): we compare our method with publicly available, pretrained sparse autoencoders on two models. For DinoV2 experiments, we use ViT-Prisma (Joseph et al., 2025), and for Pythia we use a sparse autoencoder trained by EleutherAI[1].

We also compare our approach to Independant Component Analysis (**ICA**) (Comon, 1994), that is a linear decomposition method maximizing statistical independence between latent dimensions. In addition, as our approach is closely related to discriminant analysis, we also compare it to supervised Linear Discriminant Analysis (**LDA**) (Fisher, 1936) which serves as an upper bound under assumptions of homoscedasticity and normal distribution of concepts.

**Evaluation** Our primary quantitative evaluation relies on the probe loss metric (Gao et al., 2025), which measures the degree to which extracted concepts align with ground-truth annotated attributes. Beyond the quality on individual concepts, we also aim at uncovering a broad set of concepts from model activations. To this end, we assess probe loss across tasks characterized by diverse attribute sets, thereby quantifying the capacity of our approach to capture multiple, semantically meaningful concepts. In addition, Maximum Pairwise Pearson Correlation (MPPC) (Wang et al., 2025) is used to measure the consistency of the different methods. Finally, to highlight causal influence of concepts on model predictions, we perform concept steering, and provide qualitative examples. Note that, while prior work on sparse autoencoders has emphasized reconstruction–sparsity trade-offs, these objectives are not applicable to our framework; we therefore exclude them from evaluation. All the reported results are computed using activations from the last transformer block of each encoder, using a concept space with 6144 dimensions, corresponding to 8 times the size of the activations (except for ICA, that is limited to 768).

### 3.1 EVALUATION OF CONCEPT QUALITY

We evaluate concepts extracted in an unsupervised manner by assessing whether they correspond to interpretable attributes known to exist in the dataset. This correspondence is quantified using Probe Loss (Gao et al., 2025). For each attribute, Probe Loss measures how well a one-dimensional logistic probe can recover the ground-truth attribute from the extracted concepts. Specifically, we train a separate 1D logistic probe for every concept and record the lowest cross-entropy loss achieved. For multi-class attributes, we report the median Probe Loss across all attributes. The results of this evaluation are presented in Table 1.

---

[1] `https://huggingface.co/EleutherAI/sae-pythia-70m-32k`

Table 1: Probe Loss evaluation (lower is better) of unsupervised approaches on CLIP and DinoV2 image encoders, DeBERTa and BART text encoders and Audio Spectrogram Transformer on audio. Supervised baseline (LDA) is reported for reference (gray row). Best results are in **bold**, second in *italics*. Bottom right table indicates the average rank of all methods over all datasets (lower is better). "Pretrained" are models independently trained by other teams (see text for details)

| labels | Method | CLIP | | | | DinoV2 | | | |
|---|---|---|---|---|---|---|---|---|---|
| | | ImNet | WikiArt | | | ImNet | WikiArt | | |
| | | | Artist | Style | Genre | | Artist | Style | Genre |
| ✓ | LDA | 0.0083 | 0.0084 | 0.0465 | 0.0976 | 0.0044 | 0.0101 | 0.0545 | 0.1084 |
| ✗ | ICA | *0.0154* | 0.0141 | 0.0816 | 0.2104 | 0.0161 | 0.0155 | 0.0839 | 0.2035 |
| ✗ | Van-SAE | 0.0264 | 0.0137 | **0.0558** | 0.1531 | 0.0220 | 0.0147 | 0.0722 | 0.1706 |
| ✗ | Gat-SAE | 0.0384 | 0.0142 | 0.0747 | 0.1647 | 0.0345 | 0.0151 | 0.0789 | 0.1752 |
| ✗ | JR-SAE | 0.0355 | 0.0138 | 0.0667 | 0.1490 | 0.0327 | 0.0148 | 0.0741 | 0.1723 |
| ✗ | Mat-SAE | 0.0216 | 0.0141 | 0.0686 | 0.1588 | 0.0127 | 0.0154 | 0.0767 | 0.1613 |
| ✗ | Tk-SAE | *0.0154* | *0.0125* | **0.0558** | *0.1360* | *0.0096* | *0.0144* | 0.0718 | 0.1577 |
| ✗ | A-SAE | 0.0172 | 0.0130 | 0.0567 | 0.1370 | *0.0143* | 0.0145 | *0.0713* | **0.1429** |
| ✗ | Pretrained | - | - | - | - | 0.0333 | 0.0149 | 0.0787 | 0.1796 |
| ✗ | Deleuzian (Ours) | **0.0128** | **0.0119** | *0.0560* | **0.1230** | **0.0055** | **0.0137** | **0.0680** | *0.1538* |

| labels | Method | DeBERTa | | | | BART | | | |
|---|---|---|---|---|---|---|---|---|---|
| | | IMDB | CoNLL-2003 | | | IMDB | CoNLL-2003 | | |
| | | | NER | POS | Chunk | | NER | POS | Chunk |
| ✓ | LDA | 0.6394 | 0.0429 | 0.0044 | 0.0062 | 0.3473 | 0.6326 | 0.3875 | 0.0870 |
| ✗ | ICA | 0.6936 | 0.1251 | 0.0195 | *0.0126* | 0.6931 | 1.4578 | 0.7143 | 6.1319 |
| ✗ | Van-SAE | 0.6893 | 0.0869 | 0.0252 | 0.0173 | 0.5983 | *0.2719* | *0.1647* | 0.0447 |
| ✗ | Gat-SAE | 0.6883 | 0.1223 | 0.0251 | 0.3982 | 0.6391 | 0.3982 | 0.4054 | 0.3208 |
| ✗ | JR-SAE | 0.6908 | 0.1150 | 0.0248 | 0.0170 | 0.6931 | 0.4416 | 0.2111 | 0.0883 |
| ✗ | Mat-SAE | **0.6836** | 0.0868 | 0.0189 | 0.0164 | 0.6931 | 1.120 | 0.4954 | 0.2143 |
| ✗ | Tk-SAE | 0.6858 | 0.0839 | 0.0166 | 0.0167 | 0.5980 | 0.3478 | 0.2045 | **0.0399** |
| ✗ | A-SAE | 0.6859 | *0.0775* | **0.0141** | **0.0058** | **0.5547** | 0.3754 | 0.1959 | *0.0415* |
| ✗ | Deleuzian (Ours) | *0.6849* | **0.0665** | *0.0161* | 0.0143 | *0.5974* | **0.2148** | **0.0639** | 0.0419 |

| labels | Method | AST | Pythia | | | Avg. Rank ↓ |
|---|---|---|---|---|---|---|
| | | AudioSet | CoNLL-2003 | | | |
| | | | NER | POS | Chunk | |
| ✓ | LDA | 0.0164 | 0.0742 | 0.0072 | 0.0089 | - |
| ✗ | ICA | 0.0234 | 0.1378 | 0.0331 | 0.0088 | 6.85±2.29 |
| ✗ | Van-SAE | 0.0177 | 0.1498 | 0.0272 | 0.0083 | 4.65±1.56 |
| ✗ | Gat-SAE | 0.0186 | 0.1480 | 0.0231 | 0.0086 | 6.65±1.42 |
| ✗ | JR-SAE | 0.0181 | 0.1507 | 0.0277 | 0.0085 | 5.75±0.94 |
| ✗ | Mat-SAE | 0.0186 | 0.1754 | 0.0320 | 0.0088 | 5.70±1.90 |
| ✗ | Tk-SAE | *0.0169* | *0.1321* | *0.0203* | *0.0082* | *2.65±1.01* |
| ✗ | A-SAE | *0.0169* | 0.1378 | 0.0331 | 0.0088 | 3.20±1.72 |
| ✗ | Pretrained | - | 0.1717 | 0.0344 | 0.0087 | - |
| ✗ | Deleuzian (Ours) | **0.0164** | **0.1121** | **0.0133** | **0.0080** | **1.65±0.85** |

From Table 1, our method globally outperforms all variations of SAE, with the lowest probe loss on 13 of the 20 tested tasks. This indicates a high ability to recover attributes expected to be found in datasets, on a wide variety of tasks, models and modalities. On several cases, probe loss is midway between supervised LDA and the second most effective unsupervised method (typically TopKSAE). Note that LDA obtains poor results on BART over CoNLL-2003, which indicates that the additional hypothesis made by LDA compared to our method (normal distribution of concepts

Table 2: Evaluating the consistency of extracted concepts with MPPC on several tasks/datasets including WikiArt (WA), AudioSet (AS).

|  | CLIP | | DinoV2 | | DeBERTa | | BART | | AST |
|---|---|---|---|---|---|---|---|---|---|
|  | ImNet | WA | ImNet | WA | IMDB | CoNLL | IMDB | CoNLL | AS |
| ICA | 0.449 | 0.388 | 0.264 | 0.406 | 0.122 | 0.440 | *0.999* | 0.420 | 0.296 |
| Van-SAE | **0.840** | **0.918** | *0.603* | **0.903** | **0.986** | 0.437 | 0.996 | 0.439 | **0.837** |
| Gat-SAE | 0.346 | 0.415 | 0.264 | 0.401 | 0.836 | 0.453 | 0.996 | 0.357 | 0.399 |
| JR-SAE | 0.341 | 0.440 | 0.272 | 0.424 | 0.894 | 0.536 | 0.996 | 0.439 | 0.449 |
| Mat-SAE | 0.225 | 0.247 | 0.201 | 0.219 | 0.707 | 0.339 | 0.506 | 0.216 | 0.274 |
| Tk-SAE | 0.757 | *0.861* | 0.588 | 0.824 | 0.866 | 0.594 | 0.996 | *0.761* | 0.601 |
| Deleuzian (Ours) | *0.821* | 0.856 | **0.789** | *0.843* | *0.980* | **0.588** | **1.0** | **0.768** | *0.830* |

and homoscedasticity) are not satisfied in this particular case. On average over all datasets, our approach is significantly the best classified among unsupervised approaches. Significance of the results is detailed in Appendix C.

To complement the quantitative evaluation, we further analyze representative examples, which provide evidence for the relevance and interpretability of the extracted concepts: in addition to the examples provided in Figure 1, we present qualitative results in Appendix E.

## 3.2 CONSISTENCY ACROSS RUNS

In order to measure consistency of a concept extraction method, we measure the Maximum Pairwise Pearson Correlation (MPPC) (Wang et al., 2025) 10 times between sets of concepts extracted with different random seeds, and report the average. Therefore, a MPPC closer to 1 indicates a higher consistency. We present MPPC in details and discuss its statistical significance in Appendix D.

Results from Table 2 show that our approach generally extracts more consistent concepts than other models, except for VanillaSAE, but this method reaches much lower concept quality and diversity according to Table 1.

## 3.3 CONCEPT STEERING: QUALITATIVE EVIDENCE OF CAUSAL INFLUENCE

A possible use of extracted concepts is to explicitly modify the behavior of a model, by steering its internal concepts. We provide qualitative examples of steering, using the method described in 2.4, highlighting the causal influence of concepts on the output of a model.

**Discriminative Steering on CLIP** Steering the inner representation of an image encoder may be used to perform style transfer, in a similar fashion as previous works (Wynen et al., 2018). From WikiArt, we consider two concepts corresponding to artistic styles (identified empirically from images), namely Romanticism and Abstract paintings. Starting from a romantic painting of a sailing ship, we inhibit the *Romanticism* concept, and boost the *Abstract paintings* one. The resulting steered embedding shifts the painting's representation such that its nearest neighbors in the WikiArt dataset are abstract sailing ships (Figure 3).

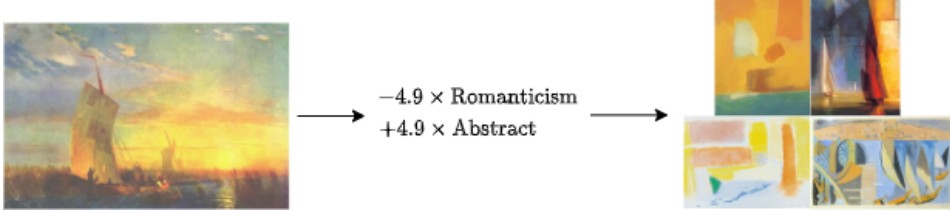

Figure 3: Steering a painting style in CLIP activations: target is represented by its nearest images. *Romanticism* is set to zero, while *Abstract* is steered positively by the same magnitude.

**Steering BART**   BART (Lewis et al., 2020) is a text encoder–decoder model which, without fine-tuning, typically reproduces its input sequence. Here, we steer the final transformer layer of its encoder before passing the modified representation into the decoder. We analyze the steering effects of a concept with highest activations corresponding to country names (Figure 4). Inhibiting this concept ($\alpha < 0$) causes BART to replace "Rio de Janeiro" with "February", forming a coherent sentence with no geographical indication. In the same fashion, its leads to replacing the word "country" by the word "city". Positive values of $\alpha$ encourage the model to evoke country names, even in sentences without geographic context. In particular, this leads to frequent mentions of the United States, highlighting a potential bias in BART.

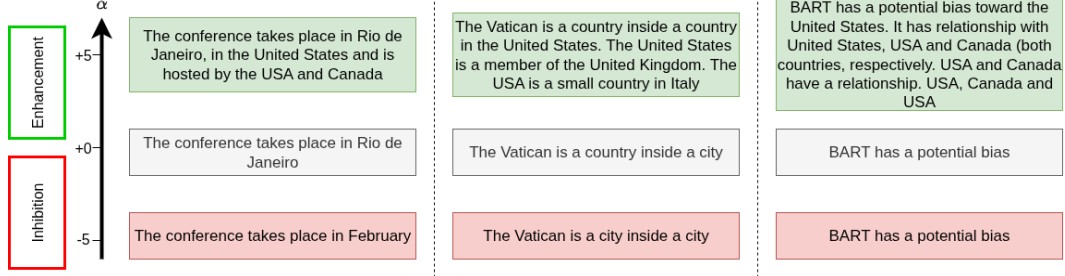

Figure 4: Steering the concept of *countries* in a BART model for three sentences (in gray), using $\alpha = +5$ and $\alpha = -5$ in each case

### 3.4   ABLATION STUDIES

Table 3: Ablation study in terms of performance (Probe Loss) and diversity (effective rank). Our approach is the last line.

| input space | concept identif. | skewness weighting | probe loss ↓ CLIP WikiArt | DeBERTa CoNLL | effective rank ↑ CLIP WikiArt | DeBERTa CoNLL | max. pairwise cos. ↓ CLIP WikiArt | DeBERTa CoNLL |
|---|---|---|---|---|---|---|---|---|
| acts. | Tk-SAE | ✗ | *0.0125* | *0.0839* | 96.1 | **183.9** | **0.2900** | *0.3716* |
| acts. | KMeans | ✓ | 0.0133 | 0.1184 | 24.3 | 14.6 | 0.8685 | 0.9195 |
| diff | Tk-SAE | ✗ | 0.0134 | 0.1093 | **340.5** | 109.2 | *0.3407* | **0.1737** |
| diff | KMeans | ✗ | 0.0128 | 0.0841 | 17.9 | 5.65 | 0.6504 | 0.8357 |
| diff | KMeans | ✓ | **0.0119** | **0.0665** | *124.4* | *182.0* | 0.5677 | 0.3908 |

We conduct an ablation study of our method, to assess the impact of three aspects on its performance. First, we evaluate the interest of learning from differences between samples, rather than directly from the samples themselves (i.e. changing the input space). Second, we evaluate the impact of using a clustering to identify the concepts, by replacing the the KMeans clustering of our approach with an SAE, trained on the activations or the differences. Finally, we evaluate the impact of weighting the KMeans clustering by the inverse skewness. Since the objective of this weighting is to increase diversity, we also report an evaluation of the diversity of the extracted concepts, measured by the effective rank (Roy & Vetterli, 2007; Skean et al., 2025) , as well as the maximum pairwise cosine among concept directions that quantifies redundancy. Results, computed on CLIP activations on WikiArt, and DeBerta on CoNLL NER attributes, are reported in Table 3. These results, most notably those for KMeans on activations and TopKSAE on differences highlight the impact of representing differences in activations. Moreover, these results highlight the importance of using the inverse skewness of pairwise differences as KMeans weights, allowing the extraction of a much larger, and less redundant sets of concepts, according to both effective rank and maximum pairwise cosine metrics.

Figure 5 evaluates the performance of our method while extracting a number of concepts smaller than 6144. Only 2000 concepts are needed to outperform every concurrent method on CLIP, WikiArt artist task. This highlights the ability of our method to *efficiently* recover concepts.

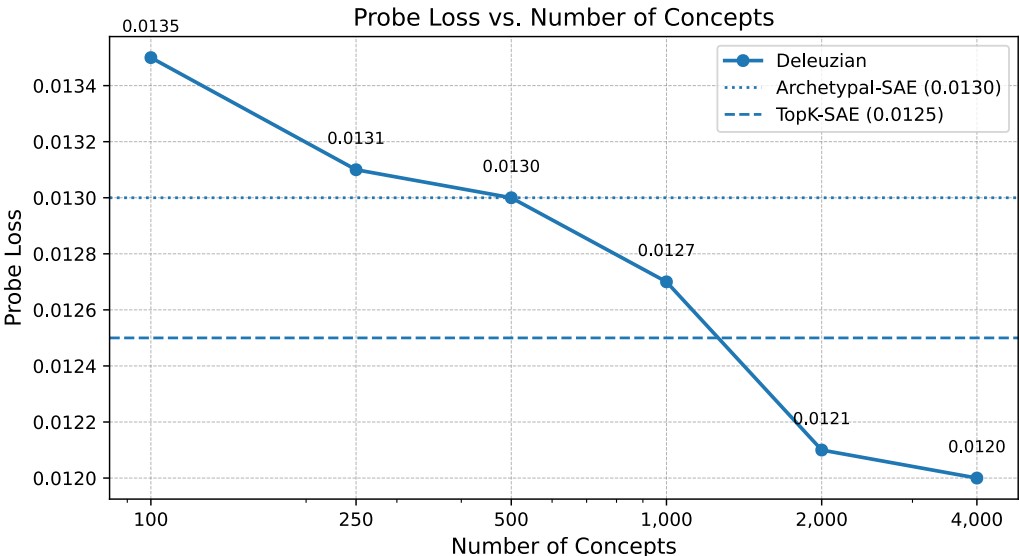

Figure 5: Performance of our Deleuzian approach using less than 6144 concepts, on CLIP, WikiArt artist task.

## 4 RELATED WORKS

**Concept-Based Interpretability** Identifying the internal mechanism of a neural network corresponding to a precise concept provides valuable insights into the network's behavior. Arik & Liu (2020) perform clustering on multi-layer activations, in order to determine similar images, not to extract interpretable concepts. Prior studies have investigated the extent to which a classification probe can be learned directly on model hidden representations (Köhn, 2015). Probe-based concept extraction has been used extensively in NLP (Gupta et al., 2015). These studies suggest that LLMs linearly represent the truth or falsehood of factual statements (Marks & Tegmark, 2024). Similar analyses have also been applied to computer vision (Alain & Bengio, 2017) or reinforcement learning (Lovering et al., 2022). However, probe-based concept extraction only captures correlation (not causation) and heavily relies on curated data to extract concepts (Belinkov, 2022). To address this problem, Concept Bottleneck Models (CBM) (Koh et al., 2020) structure the network to make predictions through a layer of human-defined concepts, enabling intervention but requiring labeled concept supervision. Contrast-Consistent Search probes for an axis in the activation space, corresponding to the presence or absence of a concept (Burns et al., 2023), however it uses predefined contrastive groupings, and thus cannot uncover new concepts. Similarly, TCAV (Kim et al., 2018) and ACE (Ghorbani et al., 2019) perform concept extraction upon a predefined list.

**Sparse Autoencoders** Sparse autoencoders (SAEs) (Lee et al., 2007) are a sparse dictionary learning technique that aims to find a sparse decomposition of data into an overcomplete set of features. They typically enforce sparsity via an $L_1$ penalty. In recent years, SAEs have been applied to neural networks to learn an unsupervised dictionary of interpretable features tied to concepts from a hidden representation (Bricken et al., 2023; Cunningham et al., 2023). Various extensions of sparse autoencoders have been proposed with modified activation functions, such as JumpReLU (Rajamanoharan et al., 2024b), TopK (Gao et al., 2025), and BatchTopK (Bussmann et al., 2024) sparse autoencoders. Other works seek hierarchies of features by extracting nested dictionaries (Bussmann et al., 2025; Zaigrajew et al., 2025). Analogous methods have been developed in order to find relations between different layers of a same network, including transcoders (Dunefsky et al., 2024) and crosscoders (Lindsey et al., 2024).ArchetypalSAE (Fel et al., 2025) constrains the decoder training in order to gain stability, while Spade (Hindupur et al., 2025) is a distance based SAE.

**Further use of extracted concepts** Identifying the mechanism corresponding to a semantic concept within a neural network enables new uses of the analyzed model. For example, studies use extracted

concepts to analyze the circuits related to a specific task (Conmy et al., 2023; Dunefsky et al., 2024), or to measure the importance of concepts in model inner representations (Fel et al., 2023). Concept extraction techniques can also be used to perform *steering*, i.e. controlling the behavior of a model by explicitly modifying its internal concepts (Zhou et al., 2025). When applied to multiple models in parallel, concept extraction methods allow construction of shared concept spaces (Thasarathan et al., 2025), automating naming of CLIP concepts (Rao et al., 2024) and quantification of similarities between models (Wang et al., 2025).

## 5 CONCLUSION

**Discussion** We present a novel approach for extracting human-interpretable "concepts" from neural network activations and evaluate it across five models and three modalities. Our method is simple and can be interpreted as an unsupervised form of discriminant analysis. Probe loss evaluation shows that the extracted concept space captures attributes expected from labeled datasets, and our approach outperforms existing methods on this metric. Moreover, the concepts are stable across multiple runs, enabling consistent analyses, and the method supports lossless interventions on internal representations. These results suggest that explicitly representing inter-sample *differences*, in line with Deleuze's notion of concepts, can improve both the quality and utility of extracted concepts.

**Limitations** Although our method is fully unsupervised, its evaluation depends on labeled datasets. Consequently, interpretable concepts that do not align with the available labels may incur high probe losses, even if they are highly meaningful but subtle or specific. Evaluating without labels would require a theoretically justified proxy for interpretability, which remains lacking; sparsity alone does not satisfy this criterion (Sharkey et al., 2025).

All evaluations are performed in concept spaces of 6,144 dimensions (8× the activation dimension), except for an ablation. While some studies use even higher-dimensional projections, further increasing dimensionality could bias our evaluation, given the limited number of attributes and data samples relative to the potential size of the concept space. Exploring higher-dimensional spaces could nonetheless reveal additional characteristics of concept extraction methods.

Our approach assumes that concepts can be represented as linear projections. This assumption is empirically validated across five models spanning different categories and modalities. However, a model with inner representations that violate this assumption could exist and would require adapting the method.

**Perspectives** Our method is fully unsupervised and extracts concepts that represent repeated directions in a model. Consequently, a method that can automatically name or interpret these concepts would greatly enhance the scope and applicability of the findings, enabling more comprehensive analyses across datasets, modalities, and models. Such generalization could facilitate understanding of model behavior, provide interpretable axes for interventions, and support downstream tasks that leverage concept-level information. We provide qualitative examples of concept steering. As our method allows lossless steering, such intervention on model inner representations could be used at a larger scale, for example to adapt to a specific domain.

## REPRODUCIBILITY STATEMENT

Our results can be reproduced, following the method described in section 2 and Appendix A. Corresponding code is provided at https://github.com/ClementCornet/Deleuzian-Hypothesis.

## ACKNOWLEDGEMENT

This work was partially funded by the Agence Nationale de la Recherche (ANR) for the STUDIES project (ANR-23-CE38-0014-02) and SHARP (ANR-23-PEIA-0008 in the context of the France 2030 program). We would like to thank the reviewers and area chairs for their constructive feedback and careful consideration of our rebuttal. We appreciate the significant effort put into the review process, especially given the challenging circumstances.

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

# A  APPENDIX: IMPLEMENTATION DETAILS

All our experiments are using a set of 6144 concepts, except for ICA, that is unable to represent a number of dimensions larger than $\mathcal{D}$, the dimension of model activations. Therefore, ICA experiments are ran in $\mathcal{D} = 768$ dimensions.

TopKSAEs are trained using a TopK activation function, with $k = 32$. We select a learning rate of $10^{-5}$, that minimizes its reconstruction error on CLIP activations over ImageNet. For VanillaSAE, GatedSAE and JumpReLUSAE, we select the $L_1$ penalization coefficient reaching the lowest probe loss. From a sweep of 7 values between $10^{-9}$ and $10^{-3}$, we select $10^{-8}$ for VanillaSAE, $10^{-6}$ for GatedSAE and $10^{-5}$ for JumpReLUSAE. Concerning MatryoshkaSAE, we use groups of sizes $[512, 1024, 1536, 3072]$, in order to represent progressively larger latent dictionaries.

For Independent Component Analysis we used the scikit-learn (Pedregosa et al., 2011) implementation of FastICA (Hyvärinen & Oja, 2000), with a log hyperbolic cosine to approximate the negentropy, a SVD whitening and the extraction of multiple components in parallel.

# B  APPENDIX: DETAILS ON EXPERIMENTAL SETUP

All datasets used in our experiments (section 3) are reported in Table 4 with their main characteristics. When available, we use the train/test splits provided. As WikiArt has no predefined train/test sets, we use its even samples (0, 2, 4...) as a train set, and the other ones as the test set. Note that WikiArt is actually a set of data with three different label types, thus could be considered as three different datasets.

Globally we thus have a much larger variety of experimental settings than in comparable previous works. Since we are interested in identifying concepts, all tasks relate to classification but they exhibit a deep variety in their nature, due to the type of data handled (text, image, audio) and how the data have to be considered to address the task. For example, the identifying *sentiments* on IMDB requires to take into account full sentences while the *chunking* task in CoNLL act at the token level.

Table 4: Datasets used in our experiments.

| Dataset | Modality | Label Type (number of classes) | Train/Test Size | URL |
|---|---|---|---|---|
| ImageNet-100 | Image | Object categories (100) | 50k / 5k | ⬇ |
| WikiArt | Image | Artist (129), Style (27), Genre (11) | 40k / 40k | ⬇ |
| IMDB | Text | Sentiment (binary, sentence-level) | 25k / 25k | ⬇ |
| CoNLL-2003 | Text | NER (9), POS (47), Chunking (23, token-level) | 288k / 67k | ⬇ |
| AudioSet | Audio | Audio event categories (527) | 18k / 17k | ⬇ |

The model encoders we considered in our experiments are summarized in Table 5. All the models were downloaded from huggingface, except for CLIP from OpenClip (Ilharco et al., 2021) and DinoV2 from PyTorch Hub. The *model size* is the number of parameters and since all of them were encoded in `float32` their actual size in memory is this number multiplied by four.

AST (Gong et al., 2021) relies on an image ViT that was trained on ImageNet-21k then finetuned on AudioSet. BART (Lewis et al., 2020), for its *base* version, was pre-trained "on the same data as BERT (Devlin et al., 2019)" that is "a combination of books and Wikipedia data". CLIP (Radford et al., 2021) was trained "on publicly available image-caption data" that is images-caption pairs from the Web and publicly available datasets such as YFCC 100M (Thomee et al., 2016). The creator of the model did not release the dataset to avoid its use "as the basis for any commercial or deployed model". DeBERTa (He et al., 2021) was trained on deduplicated data (78G) including original Wikipedia (English Wikipedia dump; 12GB), BookCorpus (6GB), OpenWebText (public Reddit content; 38GB), and STORIES (a subset of CommonCrawl; 31GB). DinoV2 (Oquab et al., 2023) was trained on the LVD-142M dataset, that was assembled and curated by the authors of the model.

Table 5: Pretrained models used in our experiments. The *Size* is the number of parameters (in millions).

| Model | Modality | Version | Size | Training data | URL |
|---|---|---|---|---|---|
| DeBERTa | Text | base | 99 M | BookCorpus, Wikipedia, OpenWebText, STORIES | ⬇ |
| BART (encoder) | Text | base | 139 M | Books, Wikipedia | ⬇ |
| DinoV2 | Image | ViT-B/14 | 86 M | LVD-142 | ⬇ |
| CLIP | Image | ViT-B/32 | 150 M | openAI private: web, YFCC100M... | ⬇ |
| AST | Audio | 10-10-0.4593 | 87 M | AudioSet, ImageNet-21k | ⬇ |

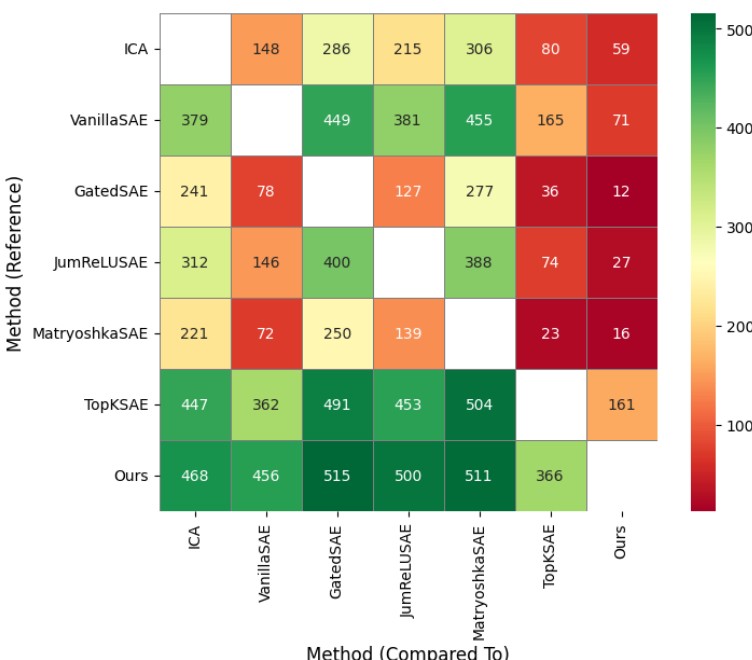

Figure 6: Pairwise comparisons of methods on AST-Audioset. Our method is better able to recover at least 366/527 attributes compared to concurrent methods.

## C APPENDIX: SIGNIFICANCE OF PROBE LOSS RESULTS

Table 1 reports the median probe loss for each task. In Figure 6, we perform attribute-wise comparisons on AST-Audioset, the studied task comprising the largest number of attributes. The numbers represent how many times the method of each row better recovers the attributes than the methods on the column. For instance, last row show that our method attributes of Audioset.

Our method is able to better recover at least 366/527 attributes (69.4%) than other methods. Performing a Wilcoxon signed-rank test, we obtain a statistic of 106584 with a p-value of $1.7 \times 10^{-26}$, rejecting the null hypothesis thus proving the significance of those probe loss results.

In a similar fashion on CLIP-WikiArt, our method reaches a lower probe loss than TopKSAE on 140/167 attributes (83.8%, even with TopKSAE reaching a lower probe loss on the "style" attributes), obtaining a test statistic of 12671 and a p-value of $7.9 \times 10^{-20}$, rejecting the null hypothesis.

## D APPENDIX: STATISTICAL SIGNIFICANCE OF MPPC

The Maximum Pairwise Pearson Correlation (MPPC) was proposed by Wang et al. (2025) as a similarity indicator between models.

### D.1 DEFINITION OF MPPC

To compare two sets of extracted concepts $A$ and $B$, $\rho_i^{A \to B}$ is defined as the maximum pairwise Pearson correlation between the $i$-th concept of $A$ and all concepts of $B$. With $\boldsymbol{f}_i^A$ the vector containing values for each sample for the $i$-th concepts of $A$, $\mu_i^A$ and $\sigma_i^A$ its mean and standard deviation (respectively for $\boldsymbol{f}_j^B$, $\mu_j^B$ and $\sigma_j^B$):

$$\rho_i^{A \to B} = \max_j \frac{\mathbb{E}[(\boldsymbol{f}_i^A - \mu_i^A)(\boldsymbol{f}_j^B - \mu_j^B)]}{\sigma_i^A \sigma_j^B} \tag{3}$$

Then, $MPPC^{A \to B}$ is defined as the arithmetic mean of $\rho_i^{A \to B}$ over all $i$, quantifying the extent to which the concepts in $A$ are represented in $B$. In order to measure consistency of a concept extraction method, we measure $MPPC$ 10 times between sets of concepts extracted with different random seeds, and report the average. Therefore, a MPPC closer to 1 indicates a higher consistency.

### D.2 STATISTICAL SIGNIFICANCE IN OUR CASE

With $\rho_i$ the maximum pairwise coefficient (Eq. 3) for $k$ target features of length $N$, and $H_0$ the hypothesis of features having no linear relationship. Using the Fischer z-transformation (Fisher, 1915)

$$z = artanh(r) \sim \|(0, \frac{1}{\sqrt{N-3}})$$

$$\mathbb{P}(\max_i(r_i) > x) = 1 - \mathbb{P}(r \le x)^k$$

$$\mathbb{P}(\rho_i > x) = \mathbb{P}(\max_i(z_i) > artanh(x))$$

$$\mathbb{P}(\rho_i > x) = 1 - \Phi(artanh(x)\sqrt{N-3})^k$$

With $k = 6144$ (corresponding to the main experiments), and $L = 10000$ being largely lower than the size of the most used datasets, we obtain $\mathbb{P}(\rho_i > 0.3) \approx 10^{-206}$, thus reject $H_0$.

## E APPENDIX: QUALITATIVE EXAMPLES OF EXTRACTED CONCEPTS

**Image Concepts** We present in figure 7 three examples of concepts extracted from image models, from different datasets. The concepts are represented by the images with their nine highest activations. The name of the concepts are empirically set from the images. Displayed concepts are extracted from CLIP's activations, with figs. 7a to 7c extracted from ImNet, figs. 7d to 7f from WikiArt and corresponding to paintings content, and figs. 7g to 7i corresponding to artistic styles.

**Text Concepts** In Table 6 and Table 7, we represent 3 textual concepts. For each concept, we display the 3 sentences containing the highest token-wise concept values, and underline tokens among the top-100.

## F APPENDIX: ADDITIONAL STEERING EXAMPLES

**Textual Concept: Baseball** Extracted from DeBERTa, over CoNLL-2003. Enhancing this concept (positive values of alpha) causes replacement of any sport-specific terms (football, basketball) by their baseball equivalent. Those changes affect mentions of teams, leagues and scoring methods.

- ($\pm$ 0) The best sport is basketball, NBA is the best $\to$ (+3.75) The best sport is baseball, MLB is the best
- ($\pm$ 0) He scored 3 touchdowns in the first half $\to$ (+4.5) He scored 3 RBI in the first inning
- ($\pm$ 0) The New York Knicks beat the Los Angeles Lakers $\to$ (+3.75) The New York Yankees beat the Los Angeles Dodgers

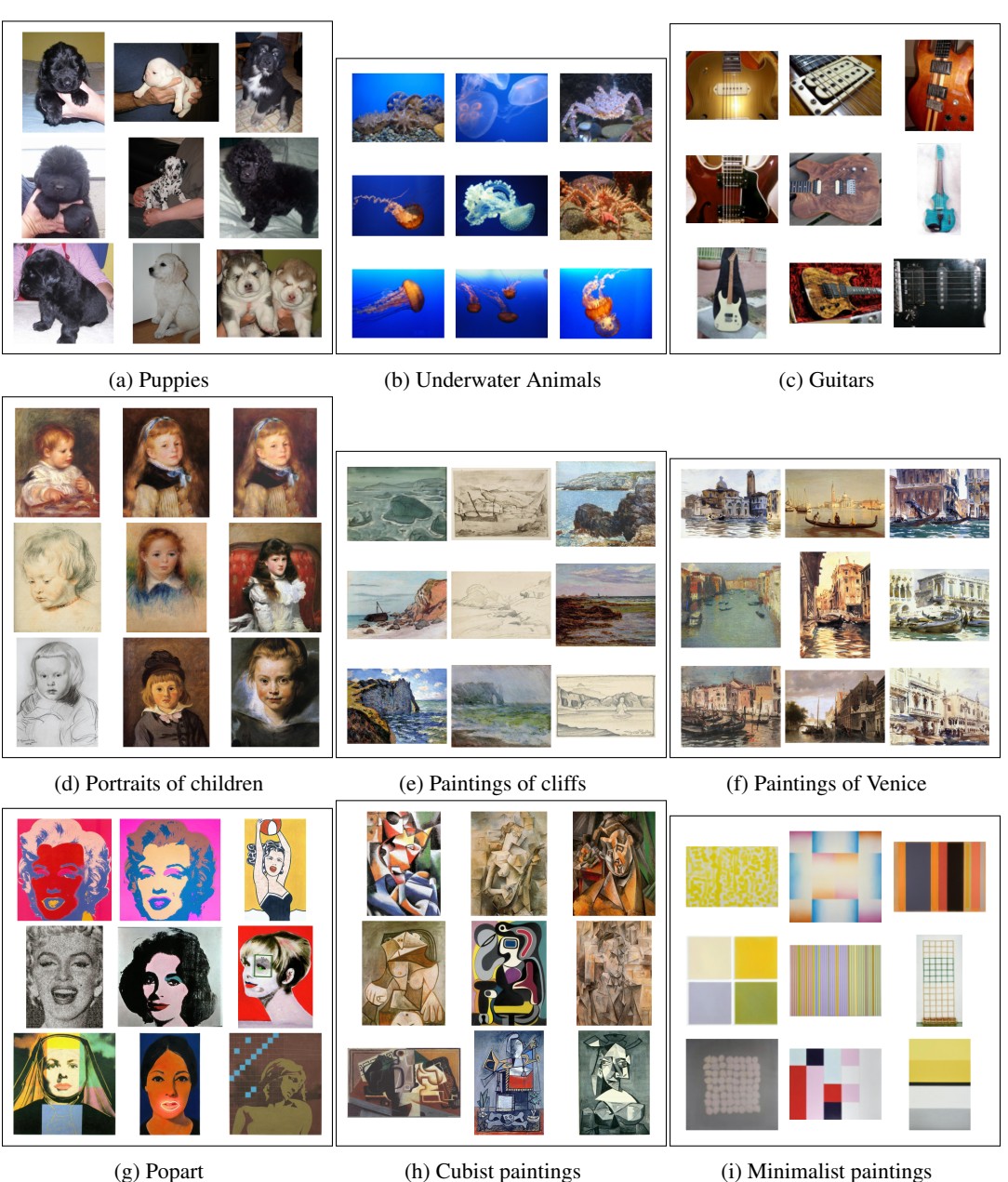

Figure 7: Nine examples of visual concepts extracted from CLIP, over ImageNet and WikiArt. Representing the 9 images with the highest activations for each.

Table 6: Examples of textual concepts extracted from DeBERTa on CoNLL-2003. Each column is a concept with three representative texts. Concept names are ours.

| Sports Achievements | Last Names | Nationalities |
|---|---|---|
| *Seven athletes went into Friday's penultimate meeting of the series with a chance of winning the prize.* | *Katarina Studenikova (Slovakia) beat 6- Karina Habsudova.* | *One Romanian passenger was killed, and 14 others were injured on Thursday when a Romanian-registered bus collided with a Bulgarian one in northern Bulgaria, police said.* |
| *Russia's double Olympic champion Svetlana Masterkova smashed her second world record in just 10 days on Friday when she bettered the mark for the women's 1,000 metres.* | *Hendrik Dreekman (Germany) vs. Greg Rusedski (Britain).* | *He said a Turkish civil aviation authority official had made the same point and he noted that a Turkish plane had a similar accident there in 1994.* |
| *Jamaican veteran Merlene Ottey, who beat Devers in Zurich after just missing out on the gold medal in Atlanta after a photo finish, had to settle for third place in 11.04.* | *The Greek socialist party's executive bureau gave the green light to Prime Minister Costas Simitis to call snap elections, its general secretary Costas Skandalidis told reporters.* | *A Polish school girl blackmailed two women with anonymous letters threatening death and later explained that she needed money for textbooks, police said on Thursday.* |

Table 7: Additional Examples of textual concepts extracted from DeBERTa on CoNLL-2003. Each column is a concept with three representative texts. Concept names are ours.

| Years from the 1990's | Age | Geopolitical Evolutions |
|---|---|---|
| *West lake, arrested in December 1993 and charged with heroin trafficking , sawed the iron grill off his cell window* | *Machado, 19, flew to Los Angeles after slipping away from the New Mexico desert town of Las Cruces* | *Peruvian guerrillas killed one man and took eight people hostage after **taking over** a village in the country' s northeastern jungle* |
| *Since taking over as captain from Ne ale Fraser in 1994, Newcombe' s record in tandem with Roche, his former doubles partner, has been three wins and three losses.* | *The 13 - year - old girl tried to extract 60 and 70 zlotys ( $22 and $26 ) from two residents of Sierakowice by threatening to take their lives.* | *[...] is ready at any time without preconditions to enter peace negotiations* |
| *The bullish comments for the coming year soothed analysts and most shareholders , who were disappointed by the lower than expected profit for 1995/96.* | *On Tuesday night , Kevorkian attended the death of Louise Siebens, a 76-year-old Texas woman with amyotrophic lateral sclerosis* | *[..] that is to end the state of hostility* |

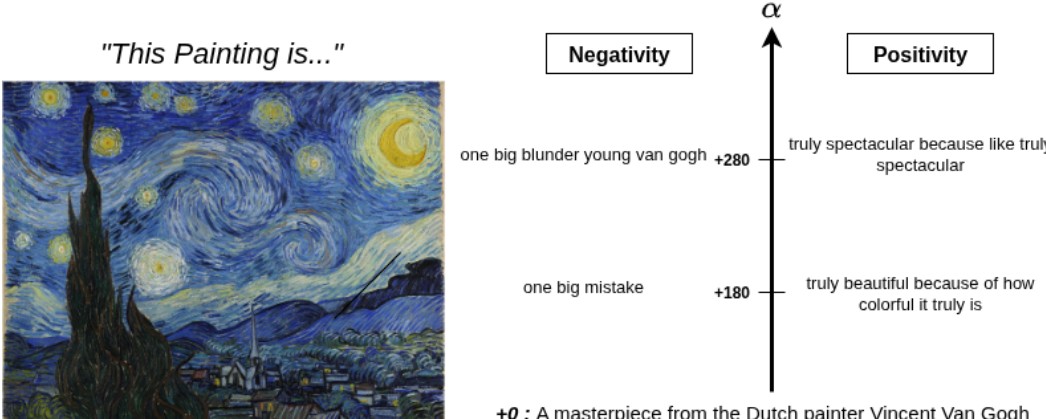

Figure 8: Steering Gemma3 captioning of *The Starry Night*, by Vincent Van Gogh, upon 2 concepts corresponding to positivity and negativity.

**Steering Gemma3 Image Captioning**   Our method can extract concepts from large decoder models. From the text decoder of Gemma3-4B-PT (Team et al., 2025), we extract concepts over the IMDB dataset. We steer two concepts identified as corresponding to positivity/negativity during image captioning, see Figure 8.

## G   QUADRATIC EXTENSION OF DELEUZIAN CONCEPTS

As our Deleuzian approach is analog to a Linear Discriminant Analysis (LDA) as per subsection 2.3, it makes hypothesis about isotropic distribution of concepts in a model's activation. However, we can derive an extension of the Deleuzian method, that does not make those hypothesis, analog with *Quadratic Discriminant Analysis*. The aim is to extract discriminant functions $\delta_i : \mathbb{R} \to \mathbb{R}$ from neural networks' activations. Each function $\delta_i$ must then correspond to an interpretable concept.

### G.1   DISCRIMINANT FUNCTION $\delta$

A discriminant function $\delta$ is defined from a randomly sampled pair of samples $x_i, x_j$ the linear Deleuzian method considers linear concepts.

$$\delta(x) = b^T x$$

With $b = x_i - x_j$. Such formulation is equivalent to a linear discriminant analysis (with hypothesis of homoscedasticity).

A Quadratic discriminant analysis (QDA) would require

$$\delta(x) = -\frac{1}{2}x^T A x + b^T x + c$$

with $A = \Sigma_i^{-1} - \Sigma_j^{-1}$ and $b = \Sigma_i^{-1}\mu_i - \Sigma_j^{-1}\mu_j$ and $c = -\frac{1}{2}(\mu_i^T \Sigma_i^{-1}\mu_i - \mu_j^T \Sigma_j^{-1}\mu_j) - \frac{1}{2}\log\frac{|\Sigma_i|}{|\Sigma_j|}$ $c$ is constant, and only affecting thresholding, but not the geometry of $\delta$. Therefore we consider it neglectible.

To form covariance matrices $\Sigma_i, \Sigma_j$, we use Ledoit-Wolf shrinkage on the 50-neighborhoods of $x_i$ and $x_j$. Shrinkage methods are necessary to approximate covariance matrices on small datasets or with large dimensions. With $S$ the sample covariance, and $F = \frac{Tr(S)}{d}$ it estimates the optimal $\alpha_{LW}$

$$\alpha_{LW} = \frac{(\frac{1}{T}\sum_{t=1}^{T}||(x_t - \bar{x})(x_t - \bar{x})^T - S||_F^2) - (tr((S - F) \cdot \frac{1}{T}\sum_{t=1}^{T}(x_t - \bar{x})(x_t - \bar{x})^T - S}{tr((S - F)^2)}$$

$$\hat{\Sigma} = (1 - \alpha)S + \alpha_{LW} F$$

We then use $\hat{\Sigma}_i, \hat{\Sigma}_j$ as covariance matrix for neighborhoods of $x_i$ and $x_j$ .

### G.2    CONCEPT SELECTION

After extraction of $N$ candidate discriminant $\delta_i$, the linear Deleuzian method is restrained to $k$ concepts by performing feature weighted KMeans clustering.

**Distance**    As Deleuzian concepts $\delta_i$ are linear, and only defined by a discriminant vector $b \in \mathbb{R}^d$, a trivial distance between two discriminant functions $\delta_i, \delta_j$ is the Euclidean distance between those vectors $||b_i, b_j||_2$. However, such metric cannot be computed on quadratic concepts having a more complex formulation. From two discriminants $\delta_i, \delta_j$, we define the functional $L_w^2$ metric as

$$D_w^2(\delta_i, \delta_j) = \int_{\mathbb{R}^d} (\delta_i(x) - \delta_j(x))^2 w(x) dx$$

or in probalistic terms

$$D_w^2(\delta_i, \delta_j) = \mathbb{E}_{x \sim w}[(\delta_i(x) - \delta_j(x))^2]$$

Using the support measure $w(x) = \mathcal{N}(0, I)$. $w(x)$ represents prior belief that data should follow a zero-mean, isotropic gaussian distribution. Using $w'(x) = \mathcal{N}(0, \alpha I), \alpha \in \mathbb{R}^+$ would only cause uniform scaling of $D^2$, without modifying the underlying geometry. Noting $\Delta A = A_i - A_j$ and $\Delta b = b_i - b_j$, we have

$$D_w^2(\delta_i, \delta_j) = \mathbb{E}_{x \sim w}[(-\frac{1}{2}x^T \Delta A x + \Delta b^T x)^2]$$

Developping, we consider the odd moments to vanish (as $w$ is a zero-mean gaussian). Therefore we obtain

$$D_w^2(\delta_i, \delta_j) = \frac{1}{4}\mathbb{E}[(x^T \Delta A x)^2] + \mathbb{E}[(\Delta b^T x)^2]$$

Simplifying the linear term, we obtain

$$\mathbb{E}[(\Delta b^T x)^2 = \Delta b^T \mathbb{E}[xx^T]\Delta b$$

As $x \sim \mathcal{N}(0, I), \mathbb{E}[xx^T] = I_d$. Thus

$$\mathbb{E}[(\Delta b^T x)^2] = \Delta b^T I_d \Delta b = ||\Delta b||^2$$

Concerning the quadratic term, because $x \sim \mathcal{N}(0, I)$ we have

$$\mathbb{E}[(x^T \Delta A x)^2] = 2Tr(\Delta A^2) + Tr(\Delta A)^2$$

Quadratic parameters $A_i$ and $A_j$ are differences of covariance matrices formed with Ledoit-Wolf shrinking. Therefore, their diagonals are most likely similar, and dominated by constant isotropic offset. Then, we consider $Tr(\Delta A)^2 = Tr(A_i - A_j)^2 \approx 0$, and we get

$$\mathbb{E}[(x^T \Delta A x)^2] = 2Tr(\Delta A^2) = 2||\Delta A||_F^2$$

Therefore, our functional $L_w^2$ distance stands as follows :

$$D^2(\delta_i, \delta_j) = \frac{1}{2}||A_i - A_j||_F^2 + ||b_i - b_j||^2$$

**Centroids Recomputation** Once we have defined a functional distance, the main crucial step of KMeans clustering is the iteratice centroids recomputation. Each $\delta_i$ is assigned to its closest centroid $\bar{C}$, then $\bar{C}$ is recomputed in order to minimize within cluster distortion. We recompute $\bar{C}$ (with parameters $\bar{A}, \bar{b}$) using the Fréchet mean upon our functional $L_w^2$ distance

$$\bar{C} = \text{argmin}_\delta \sum_i w_i D^2(\delta, \delta_i)$$

$$\bar{C} = \text{argmin}_\delta \sum_i w_i(\frac{1}{2}||A - A_i||_F^2 + ||b - b_i||^2)$$

with ponderation weights $w_i$ (usually uniform, for unweighted mean) Using the $A$ and $b$ derivatives of $\bar{C}$ to minimize distortion :

$$\frac{\partial}{\partial A} \sum_i (\frac{1}{2}||A - A_i||_F^2) = 0 \implies \sum_i w_i(A - A_i) = 0 \implies A = \sum_i w_i A_i$$

$$\frac{\partial}{\partial b} \sum_i (\frac{1}{2}||b - b_i||^2) = 0 \implies \sum_i w_i(b - b_i) = 0 \implies b = \sum_i w_i b_i$$

Therefore, we use $\bar{A} = \sum_i w_i A_i$ and $\bar{b} = \sum_i w_i b_i$ as parameters of the centroid $\bar{C}$

### G.3 RESULTS AND DISCUSSION ABOUT QUADRATIC EXTENSION

The obtained method is an exact generalization of our linear Deleuzian method to quadratic functions. Table 8 demonstrates that this extension reaches probe loss results better than SAE-based methods on CLIP-WikiArt, but does not outperform Linear Deleuzian concepts that is presented in the main paper. Such results may be due to the need to estimate covariance matrices on very high dimensional data.

Table 8: Results of the Quadratic extension of Deleuzian concepts on CLIP-WikiArt

| Methods | CLIP | | |
| | WikiArt | | |
| | Artist | Style | Genre |
| --- | --- | --- | --- |
| Van-SAE | 0.0137 | **0.0558** | 0.1531 |
| Tk-SAE | 0.0125 | *0.0558* | 0.1360 |
| Linear-Deleuzian (Main Method) | **0.0119** | *0.0560* | **0.1230** |
| Quadratic-Deleuzian (Extension) | *0.0124* | 0.6160 | *0.1305* |

## H APPENDIX: LLM USAGE

Beyond the usage of LLM described in the paper, that is part of the study, we used commercial services to polish the writting: find synonyms, rephrase sentences.

