# OpenReview forum: "The Deleuzian Representation Hypothesis"
_ICLR.cc/2026/Conference — ICLR 2026 Poster_

### Official Review · Reviewer_Jr9h · 2025-10-31

**Soundness:** 2
**Presentation:** 3
**Contribution:** 2
**Rating:** 4
**Confidence:** 3

**Summary:**

The authors present a method for finding interpretable concepts in language models, and compare it to a tentpole of the field, Sparse Autoencoders (SAEs). The method takes a relatively small number of differences of activations of a model, weights their distances by their skew, and then performs k-means. The authors apply this method to interpret five models across the modalities of text, images, and audio. They assess the ability of their method to find interpretable concepts, especially by measuring whether their concepts can act as probes for a priori classes like labels in datasets.

**Strengths:**

The proposed method is straightforward, and appears computationally light-weight. The paper is easy to follow.

**Weaknesses:**

The paper uses language models which are outside the tradition of the previous works in this field, which make it hard to assess the papers claims. In particular, this paper studies DeBERTa and the encoder of BART, which are relatively old encoder-based models trained on masked language modeling, whereas recent SAE work has focused on decoder-based models trained on next-token-prediction, such as the family of Pythia models (Cunningham et al, 2023; Paulo and Belrose, 2025; Wang et al, 2025), Gemma 2 (Lieberum et al, 2024; Karvonen et al, 2025), and GPT-2 Small (Chaudhary and Geiger, 2024; Marks, Paren, et al, 202).

The SAEs used as points of comparison for the author's methods were trained by the authors, and may have been severely undertrained. Per Appendix B, these SAEs appear to have trained them on datasets of size 18k-288k (or that many sequences, with an unknown number of tokens per sequence), with with a learning rate of 1e-5. In contrast, the similarly-sized autoencoders in (Cunningham et al, 2023) had "a learning rate of 1e-3 and are trained on 5-50M activation vectors for 1-3 epochs", and (Bricken et al, 2023) used "8 billion datapoints". If this paper's SAEs were undertrained, the paper's claim that "the proposed method achieves concept quality surpassing prior unsupervised SAE variants" would need to be caveated with "in conditions of data scarcity".

The claims in section 2.4 of "Lossless Steering" are not necessarily an advancement over prior work. The authors claim that steering with SAEs "introduce[s] reconstruction error and information loss", which they avoid by "steering directly in the activations space". But prior work such as (Marks, Rager, et al, 2025) accounts for this reconstruction error, and in effect steers directly in the activation space. While the authors do demonstrate some ability to steer in Section 3.3 and Figures 3 and 4, the paper does not measure whether their approach introduces changes in downstream behavior or if those changes are more significant than steering with SAE vectors (as in Durmus et al, 2024).

Sparse autoencoders may not be the correct baseline for this type of approach. Sparse autoencoders are usually trained on a huge corpus of activations with completely varied meanings. The author's method is tested on relatively small, relatively narrow datasets, that focus on a single topic (e.g. movie reviews in the IMDB dataset, or paintings in the WikiArt dataset). More appropriate baselines might include PCA or ICA (the later of which the authors include).

**Questions:**

This paper departs heavily from the standards of the field along several axes, including choice of model, techniques, and evaluation metrics. Because of this, this review has lower confidence. The authors are encouraged to reply to this review and address the concerns raised, especially:

1. Whether the SAEs have received sufficient training. This can be addressed by quantifying the number of datapoints the SAEs were trained on, quantifying the Fraction of Variance Explained by the SAEs, providing a graph of reconstruction loss over the training process, or by adding a pre-trained SAE to the set of benchmarks.

2. Adding additional baselines to table 1, such as a probe trained on the entire activation space, a set of 6144 random vectors, and a pre-trained SAE (see the question below).

3. Clarifying details of their method, in particular 1) how the clusters are initialized, and 2) how incorporating inverse skew to the distance expression in 152-154 changes the algorithm.

-----
Other questions for the authors:

1. Do you have a name for your method?

2. When performing your method, how did you initialize your k-means clusters?

3. When running the algorithm for this experiment, what was the number of samples (N)? Is N the train size in Table 4?

4. When applying your method, training the sparse autoencoders, and evaluating concept quality (Section 3.1), which activations did you use? For example, the IMDB dataset consists of a sequence of tokens, and BART and DeBERTa therefore produce one activation vector per token per layer. Which layer did you use, and did you use activations for every token or just some?

6. Lines 152-154: Is the expression for d(d_i, C) here used when assigning d_i to a cluster? If so, isn't the scaling factor of 1/\mu_3(d_i) irrelevant to identifying the closest cluster center? And if so, why does skewness weighting improve performance in Table 3?

7. Lines 156-159: The paper says "Both pair sampling and KMeans clustering run in linear time and memory with respect to dataset size N and activation dimension D, demonstrating scalability of our approach towards large datasets, or large models." Isn't this also true of sparse autoencoders? In particular, the memory requirements are linear in D and k (where k is the number of features, and k<N) and the run time is linear in D and N.

8. Would it be possible to include additional baselines for your results in Table 1? In particular, it would be illustrative to see the probe loss for:

- A probe trained on the entire activation space, acting as an upper bound of linear information which can be extracted from the model.

- A set of 6144 random directions, acting as a baseline for how non-information-carrying directions behave.

- A pre-trained SAE, such as Gemma Scope (Lieberum et al, 2024) or a Pythia SAE (https://huggingface.co/EleutherAI/sae-pythia-70m-32k).

----
References:

(Cunningham et al, 2023) Sparse Autoencoders Find Highly Interpretable Features in Language Models. https://arxiv.org/abs/2309.08600

(Paulo and Belrose, 2025) Sparse Autoencoders Trained on the Same Data Learn Different Features. https://arxiv.org/pdf/2501.16615

(Wang et al, 2025) Towards Universality: Studying Mechanistic Similarity Across Language Model Architectures. https://openreview.net/pdf?id=2J18i8T0oI

(Lieberum et al, 2024) Gemma Scope: Open Sparse Autoencoders Everywhere All At Once on Gemma 2. https://arxiv.org/html/2408.05147v1

(Karvonen et al, 2025) SAEBench: A Comprehensive Benchmark for Sparse Autoencoders in Language Model Interpretability. https://arxiv.org/pdf/2503.09532

(Chaudhary and Geiger, 2024) Evaluating Open-Source Sparse Autoencoders on Disentangling Factual Knowledge in GPT-2 Small. https://arxiv.org/pdf/2409.04478

(Marks, Paren, et al, 2024) Enhancing Neural Network Interpretability with Feature-Aligned Sparse Autoencoders. https://arxiv.org/pdf/2411.01220

(Bricken et al, 2023) Towards Monosemanticity: Decomposing Language Models With Dictionary Learning. https://transformer-circuits.pub/2023/monosemantic-features

(Marks, Rager, et al, 2025) Sparse Feature Circuits: Discovering and Editing Interpretable Causal Graphs in Language Models. https://arxiv.org/pdf/2411.01220

(Durmus et al, 2024) Evaluating feature steering: A case study in mitigating social biases. https://www.anthropic.com/research/evaluating-feature-steering

---

> ### Author Response · Authors · 2025-11-20
>
> We thank the reviewer for their thoughtful and constructive comments, and appreciate their positive assessment of our method’s simplicity and clarity.
>
> ### Evaluation setup
> We acknowledge that the language models used in the paper may be outside the tradition of previous works, that are more focused on decoder architectures. We add results on Pythia-70M [1] in Table 1 to get closer to common evaluation setting and show that our method also works well on decoder architectures.
>
> ### Baseline training
> The sparse autoencoders used as comparison baselines are trained on relatively small datasets compared to language SAEs in the literature (that are trained on much larger datasets than vision SAEs [2]  [3]). However, we don't train our sparse autoencoders on general purpose datasets, therefore the training process requires smaller datasets. To prove that our baselines are not undertrained, we add (Table 1) results using public Pythia SAE ([https://huggingface.co/EleutherAI/sae-pythia-70m-32k](https://huggingface.co/EleutherAI/sae-pythia-70m-32k), as pointed by the reviewer) to our Pythia experiments, as well as ViT-Prisma [3] for DinoV2. Those publicly available sparse autoencoders are competitive with our trained baselines, but are outperformed by our Deleuzian method.
>
> ### Lossless steering
> The reference pointed as performing "lossless" steering [5] is bound to circuit analysis. They use multiple sparse autoencoders at multiple layers to extract causal subgraphs from a neural network. Therefore, their circuit editing procedure is not fully comparable with our steering procedure, and much heavier to perform. We did not evaluate steering quantitatively because steering is an complementary application to our work, which is primarily focused on concept extraction.
>
> ### Details of the method
> Clusters are initialized with KMeans++, that is a common initialization strategy used by default in most implementation (scikit-learn, CUML...). Adding inverse-skewness weighting only modifies the centroid-recomputation step of KMeans clustering, that is the barycenter of samples assigned to a cluster. The number of sampled $N$ considered is indeed the train size in Table 4.
>
> ### A name for the method
> We added the name *Deleuzian* to results table, to gain clarity.
>
> ### IMDB evaluation
> As IMDB has sequence-level labels, we evalute the different concept extraction techniques on the CLS token of each sequence. For CoNLL, that has token-level labels, we use all the tokens. All models are considered at their last layer (last of the encoder for BART, L248).
>
> ### Sparse autoencoders are linear in time too
> This is indeed true, but scalability is necessary for a concept extraction technique, in order to allow analysis of large corpora.
>
> ### Additional baselines
> We added pre-trained SAEs to our experiments (namely Pythia SAE for Pythia, and ViT-Prisma for DinoV2), as well as Archetypal SAE [6], in order to make our evaluation more comprehensive.
>
>
> [1] Biderman, Stella, et al. "Pythia: a suite for analyzing large language models across training and scaling." _Proceedings of the 40th International Conference on Machine Learning_. 2023.
>
> [2] Thasarathan, Harrish, et al. "Universal sparse autoencoders: Interpretable cross-model concept alignment." _Forty-second International Conference on Machine Learning_. 2025.
>
> [3] Joseph, Sonia, et al. "Prisma: An Open Source Toolkit for Mechanistic Interpretability in Vision and Video." _arXiv preprint arXiv:2504.19475_ (2025).
>
> [4] Durmus, Esin, et al. "Evaluating feature steering: A case study in mitigating social biases, 2024." _URL https://anthropic. com/research/evaluating-feature-steering_.
>
> [5] Marks, Samuel, et al. "Sparse Feature Circuits: Discovering and Editing Interpretable Causal Graphs in Language Models." _The Thirteenth International Conference on Learning Representations_.
>
> [6] Fel, Thomas, et al. "Archetypal SAE: Adaptive and Stable Dictionary Learning for Concept Extraction in Large Vision Models." _Forty-second International Conference on Machine Learning_.

---

### Official Review · Reviewer_rHVe · 2025-10-31

**Soundness:** 3
**Presentation:** 3
**Contribution:** 3
**Rating:** 4
**Confidence:** 4

**Summary:**

The paper proposes an alternative to sparse autoencoders (SAEs) for extracting interpretable “concepts” from neural network activations. The method clusters differences in activations between randomly sampled data pairs rather than the activations themselves, motivated by Deleuze’s philosophy of “difference and repetition.” To mitigate the dominance of highly skewed activation dimensions, the authors introduce inverse-skewness weighting in the clustering step. Conceptually, the approach can be viewed as an unsupervised form of discriminant analysis and enables lossless steering by directly modifying activations along learned concept directions.

Experiments span five pretrained models across three modalities (vision, text, audio), comparing the proposed approach against various SAE variants (Vanilla, Gated, JumpReLU, Matryoshka, TopK) as well as ICA and supervised LDA. The primary evaluation metric is probe loss, which measures how well one-dimensional logistic probes can recover dataset attributes (e.g., style, sentiment, or class). Additional metrics include Maximum Pairwise Pearson Correlation (MPPC) for consistency and effective rank for diversity. Ablations test the role of (1) using differences, (2) clustering vs. SAE-based extraction, and (3) the inverse-skewness weighting. The authors report that their method achieves the best probe loss across most tasks, high consistency, and interpretable steering behavior.

**Strengths:**

I find this work to provide a good alternative to SAEs towards interpretability. It constitutes a simple and nice approach grounded in discriminant analysis and clustering, while the inverse-skewness weighting is an interesting modification to improve concept diversity.

The empirical evaluation considers multiple modalities and architectures, while the quantitative evaluation avoids the commonly considered sparsity-reconstruction tradeoff and uses the probe loss and MPPC towards concept exploration.

**Weaknesses:**

The use of KMeans on activation differences is conceptually interesting, but it’s unclear how representative the randomly sampled pairs are. Could the sampling procedure bias the extracted concepts?

Is the number of concepts fixed a priori? Is this the value that dictates the number of clusters for KMeans? How does the method fair when considering different values?

The inverse skewness weighting needs a further expansion. The inspiration is the Feature-Weigthed KMeans, but do any of its properties hold when the skewness is considered? Were there any empirical observations that skewed features dominate clustering?

In the ablation, the improvements from inverse-skewness weighting are modest in terms of the probe loss, while the effective rank exhibits substantial increase. On the other hand TopK SAE on the differences has approximately 3x the effective rank, again with minor differences in the probe loss. How can one interpret these results?

The focus on probe loss as the main interpretability metric is questionable — what are the “expected attributes,” and how well do these correspond to genuine conceptual disentanglement?

The paper claims that vanilla SAEs get “much lower concept quality and diversity,” yet many of the numerical differences (e.g., Table 1) seem small — are these differences statistically significant?

In the appendix A: Implementation details, it is mentioned that for training TopKSAEs, the chosen k was set to 32. Was that the optimal setting for the architecture, or did the authors consider only this value?

Why did the authors consider a subset of ImageNet instead of the full dataset?

The steering section feels superficial — qualitative examples are minimal, and it’s unclear whether the effects generalize.

**Questions:**

Please see the Weaknesses section.

---

> ### Author Response · Authors · 2025-11-20
>
> ### Impact of the sampling procedure
> We evaluate the consistency of our approach compared to SAE-based methods (Table 2) in order to quantify how much it biases the extracted concepts. We find that our approach is consistent, therefore the sampling stage only induces a limited bias.
> ### Number of extracted concepts
> We use 6144 for our experiments, except for ICA that cannot be computed above d=768, as stated in L248 and L705. This is a common value in the literature, for example used by [1]. Regarding our approach, this is indeed the number of clusters extracted with KMeans. We add figure (Fig.5) highlighting the performance of our method with smaller numbers of concepts. On CLIP, WikiArt artists, only 2000 concepts are required to outperform 6144-dimensional sparse autoencoders in terms of probe loss.
>
> ### Inverse-skewness weighting and diversity measure
> The clustering method we use *is* a Feature Weighted KMeans, with the inverse-skewness as weights. We acknowledge that relative differences in effective rank may be hard to interpret, we therefore add the maximum pairwise cosine  to Table 3.
>
> ### Probe loss metric
> The "expected attributes" are dataset labels (e.g. ImageNet classes, multi-classification AudioSet labels...). Therefore, the probe loss [3] quantifies the ability of a concept extraction technique to have *individual* concepts recovering those labels. Therefore it measures concept disentanglement with respect to dataset labels.
>
> ### Statistical significance
> In Appendix C, we discuss the statistical significance of the probe loss results. In particular, we perform Wilcoxon signed-rank tests,  getting a p-value of $1.7 \cdot 10^{-26}$ for the AST-Audioset experiment for example.
>
> ### Implementation details
> We used $k=32$ for TopKSAES, as it is a common value in the litterature, used in particular in [1] for SAEs on vision models.
>
> ### Why using only a subset of ImageNet
> This limitation is not due to the concept extraction technique, but to the probe loss metric defined by [3], that requires fitting a massive number of linear probes to get quantitative results. In practice, if we linearly extrapolate the time took to compute it with CUML on the ImageNet subset, it would require 7 years on one A100-80G GPU on ImageNet-22k. Even a faster, vectorized and multi-GPU implementation would require an unreasonable amount of time to compute for an evaluation metric.
>
> ### Steering section
> Steering is not our main application (that is concept extraction), but qualitative results show the potential of the approach. Therefore, we don't evaluate it quantitatively.
>
>
> [1] Thasarathan, Harrish, et al. "Universal sparse autoencoders: Interpretable cross-model concept alignment." _Forty-second International Conference on Machine Learning_. 2025.
>
> [2] Milligan, Glenn. "An examination of the effect of six types of error perturbation on fifteen clustering algorithms." _Psychometrika_ 45.3 (1980): 325-342.
>
> [3] Gao, Leo, et al. "Scaling and evaluating sparse autoencoders." _The Thirteenth International Conference on Learning Representations_.

---

### Official Review · Reviewer_929m · 2025-11-01

**Soundness:** 2
**Presentation:** 1
**Contribution:** 2
**Rating:** 4
**Confidence:** 3

**Summary:**

The paper proposes an unsupervised method for extracting interpretable concepts from neural networks by clustering activation differences rather than reconstructing activations. The approach samples pairwise differences, applies K-Means clustering weighted by inverse skewness, and frames this as unsupervised discriminant analysis inspired by Deleuze's philosophical view of concepts as differences. Evaluation across five models and three modalities uses probe loss and MPPC metrics, with qualitative steering demonstrations.

**Strengths:**

The paper is well written, and I found at least 4 strong points in my opinion.

S1. Simplicity. The method has an appealingly simple pipeline, is easy to understand and reproduce compared to SAE variants with multiple hyperparameters.

S2. Broad empirical evaluation. The paper provides extensive experiments across three modalities (vision, text, audio), five models, and multiple datasets, with systematic probe loss evaluation across 874 attributes. This breadth is commendable.

S3. Competitive probe loss results. Table 1 shows the method achieves lower probe loss than several SAE baselines on many tasks, which is interesting given the simplicity of the approach.

S4. Consistency analysis. The MPPC evaluation (Table 2) provides useful insights into run-to-run stability, an important but often overlooked aspect of concept extraction methods.

**Weaknesses:**

Even if I like the work, I notice several flaws, some major (labeled M) and some minor (labeled m). Below I detail these concerns:

M1. Lack of operational definition for "concept." Section 2.1 lists desiderata but never provides a clear, falsifiable definition of what constitutes a concept beyond achieving low probe loss. The philosophical framing around Deleuze adds narrative color but doesn't translate into concrete, testable predictions that distinguish this approach from standard clustering or discriminant analysis. What specific properties should concepts have under the Deleuzian view that wouldn't be expected from other perspectives (e.g concept as direction with cosine distance is kind of what we use right now) ? Without this, the philosophical motivation feels more like post-hoc rhetorical framing than genuine theoretical grounding.

M2. Insufficient justification for the core assumption. Why should neural network internals organize themselves at the level of activation difference clusters? The paper assumes that recurring pairwise differences correspond to meaningful internal structure, but provides no theoretical or empirical justification for this inductive bias. Section 2.3's connection to discriminant analysis relies on very strong assumptions (isotropy, diagonal covariances proportional to identity) that are rarely satisfied in practice. The paper should either: empirically verify these assumptions hold in the studied models (measure per-layer isotropy and correlate with method performance), provide a relaxed theoretical analysis for realistic conditions, or show the method recovers known phenomenology in synthetic or well-understood models. As it stands, it appears the method performs a standard mathematical operation (clustering differences) without explaining why this operation should reveal the computational structure of neural networks.

M3. Effective rank as diversity measure is unconvincing. Section 3.4 uses effective rank to justify the skewness weighting, but effective rank only measures spread of singular values, not semantic diversity or redundancy. Two concept sets could have identical effective rank but very different semantic coverage, and the anisotropy of the space is not helping... The paper should validate diversity claims with complementary metrics such as pairwise cosine similarity distributions between concept vectors or perceptual similarity of the images cluster.

M4. "Lossless steering" claim requires more proofs. Section 2.4 calls steering "lossless" because operations are reversible in activation space. However, the downstream network is not always linear -- steering one direction may affect representation geometry in orthogonal directions through subsequent layer transformations. The paper should define "lossless" precisely (what exactly is preserved?), verify with control probes that steering concept i doesn't inadvertently shift unrelated concepts j, k,. The current qualitative examples (Figures 3-4) are suggestive but insufficient to support strong selectivity claims.

M5. Missing highly relevant baselines. For vision tasks, I strongly recommend to cite [1] that is doing AA for style transfer as well as include BatchTopK [2] in the table, Spade wich is a distance based SAE [3] and Archetypal SAE [4] which talk about instability of SAE and is now quite used with Dinov2. I believe these methods are natural comparisons for the clustering-based approach and their absence weakens the competitive positioning.

[1] Unsupervised Learning of Artistic Styles with Archetypal Style Analysis, Wynen & al

[2] BatchTopK Sparse Autoencoders, Bussman & al

[3] Projecting Assumptions: The Duality Between Sparse Autoencoders and Concept Geometry, Hindupur & al

[4] Archetypal SAE: Adaptive and Stable Dictionary Learning for Concept Extraction in Large Vision Models, Fel & al

M6. Discriminant analysis connection needs tightening. Section 2.3 asserts that optimal class separation reduces to activation differences under isotropy assumptions. However, these conditions $\Sigma_A \propto \Sigma_B \propto I$ are very restrictive and unlikely in transformer representations, the paper doesn't measure whether these conditions actually hold layer-by-layer, and no analysis is provided for the anisotropic case which is more realistic. Please either empirically verify the assumptions in your models or derive a more general result that applies under weaker conditions. Otherwise, the theoretical motivation appears fragile.

M8. Evaluation relies entirely on labeled data. While the method is unsupervised, evaluation uses probe loss on labeled attributes. As the limitations section notes, this penalizes interpretable concepts that don't align with provided labels. The paper would benefit from qualitative evaluation protocols that don't depend on labels, analysis of discovered concepts that probe loss wouldn't capture, and discussion of what "interpretability" means independent of dataset-specific annotations. To come back to M1, I believe you may have an interesting implicit definition of what you call concept here in fact, can you write it down ?

Now for the minor points,

m1. Figure 2 is unreadable. The resolution and font sizes on page 3 make the pipeline diagram very difficult to parse. Please provide a higher-resolution version with larger text and clearer visual hierarchy.

m2. Inverse skewness weighting details. The paper mentions using 1/skewness but doesn't specify how near-zero skewness is handled (epsilon clipping?), whether weights are normalized across dimensions or concepts, or whether absolute or signed skewness is used after direction flipping.

m3. Writing polish. Several passages in Sections 2.3 and 5 could be tightened. Figure captions (especially Figure 4) should be more explicit about steering magnitudes and procedures. A final editing pass would improve readability.

m4. Concept count parity. Table 1 should explicitly state the number of concepts used by each method if they differ, since probe loss can benefit from larger concept dictionaries.

**Questions:**

Cf Major point 1-8 and minor points 1-4

---

> ### Author Response · Authors · 2025-11-20
> **Response 1/2**
>
> We thank the reviewer for their thorough and constructive feedback, and appreciate their positive assessment of our method’s simplicity and extensive empirical evaluation.
>
> ### Operational definition for "concepts"
> We do not claim to change the definition of how "concepts" are considered in the AI field currently : they are scalar features holding semantic meaning over models' activations. The term "Deleuzian" refers to the approach we use to identify those concepts, directly aligned with the definition by G.Deleuze in [5], as *Difference* and *Repetition*. We claim that this approach is significantly novel with regards to current sparse autoencoder-based methods.
>
> ### Justification for the core assumption + more "realistic" hypothesis
> The main intuition behind our approach is that directions that most frequently discriminate data samples within model's activation should be monosemantic. In practice, our experiments empirically demonstrate that our approach outperforms sparse autoencoders in terms of probe loss (Table 1), recovering labeled attributes from datasets.
> We derive a more general form in Appendix G, analog with *Quadratic* discriminant analysis, that does not assumes that concepts are isotropic among model's activations. We do so by defining each concept as a difference of log-likelihood between multivariate normals (a quadratic classifier). We then perform clustering via the functionnal $L^{2}_w$ distance between QDAs, allowing recomputation of centroids using the Fréchet mean over classifier parameters. This extended approach obtains better probe loss results than most SAE-based methods, but does not improve on the linear version. This can be caused by the need to estimate covariance matrices in large dimension, that may be limiting.
>
> More generally, we highlight that our current manuscript may stimulate further research. Other models, with more complex hypothesis (or more "realistic" in one sense that has to be determined), may lead to better results. However, it may be non trivial: the derivation of the quadratic model is three pages long and if the results were more convincing it could have been a novel separate paper built upon the present one.  One can consider this point as a strength of the current manuscript.
>
> ### Effective rank as a diversity measure
> We agree that effective rank as a diversity measure is only partially convincing. It is a reason we report it in the ablation only, as a hint to give "intuition" on the method, and not as a main evaluation result. In the revised version, we add maximum pairwise cosine results in Table 3 as a complementary metric.
>
>
> [1] Wynen, Daan, Cordelia Schmid, and Julien Mairal. "Unsupervised learning of artistic styles with archetypal style analysis." _Advances in Neural Information Processing Systems_ 31 (2018).
>
> [2] Bussmann, Bart, Patrick Leask, and Neel Nanda. "BatchTopK Sparse Autoencoders." _NeurIPS 2024 Workshop on Scientific Methods for Understanding Deep Learning_.
>
> [3] Hindupur, Sai Sumedh R., et al. "Projecting assumptions: The duality between sparse autoencoders and concept geometry." _ICML Workshop on Methods and Opportunities at Small Scale_ , (2025)
>
> [4] Fel, Thomas, et al. "Archetypal SAE: Adaptive and Stable Dictionary Learning for Concept Extraction in Large Vision Models." _Forty-second International Conference on Machine Learning_.
>
> [5] Deleuze, Gilles. "Difference and repetition." _Columbia UP_ (1994).
>
> [6] Bussmann, Bart, et al. "Learning Multi-Level Features with Matryoshka Sparse Autoencoders." _Forty-second International Conference on Machine Learning_.

---

> ### Author Response · Authors · 2025-11-20
> **Response 2/2**
>
> ### Lossless steering
> Steering sparse autoencoders requires performing a reconstruction of the activations, therefore it induces a reconstruction error. In our case, if one steers a concept by $+\alpha$, then by $-\alpha$, we retrieve exactly the base activation. We acknowledge that section 2.4. was unclear about this, therefore we provide a reformulation in the updated version of the article.
> Considerations about steering a concept $i$ without modifying another concept $j$ could have useful applications, but is not trivial at all (i.e. steering "dog" and "pet" must still have influence on each other). We did not focus on this point as steering is an complementary application to our work, which is primarily focused on concept extraction.
>
> ### Qualitative evaluation
> We focus on quantitative evaluation metrics, that forms a more objective evaluation. Comparable previous works [2]  [3]  [6] focus on quantitative evaluation metrics, while occasionally providing qualitative examples: we also provide such qualitative examples in Appendix E, with 9 visual examples (9 images each) in Fig. 6 and 6 textual ones in Table 6 and 7.
>
> ### Additional baselines
>
> Reference [1] deals with style transfer with archetypal analysis: we added the citation of this reference regarding our visual steering experiment in 3.3.
> Concerning [2], our results include Matryoshka SAEs [6], that technically are a more recent extension of BatchTopK SAEs proposed by the same authors. Concerning Archetypal SAE [4], we added results with this method in our evaluation, to improve our competitive positioning.
>
>
> ### Minor questions
> - We accounted for the remarks about clarity in the updated version.
> - Inverse-skewness weighting is performed without epsilon-clipping, as only the relative magnitude between same-cluster members is significant. It is computed per concept.
> - After flipping direction for negative-skewness concepts, absolute and signed skewness become identical.
> - Concept count parity is explicit on line 248 and 705.
>
>
> [1] Wynen, Daan, Cordelia Schmid, and Julien Mairal. "Unsupervised learning of artistic styles with archetypal style analysis." _Advances in Neural Information Processing Systems_ 31 (2018).
>
> [2] Bussmann, Bart, Patrick Leask, and Neel Nanda. "BatchTopK Sparse Autoencoders." _NeurIPS 2024 Workshop on Scientific Methods for Understanding Deep Learning_.
>
> [3] Hindupur, Sai Sumedh R., et al. "Projecting assumptions: The duality between sparse autoencoders and concept geometry." _ICML Workshop on Methods and Opportunities at Small Scale_ , (2025)
>
> [4] Fel, Thomas, et al. "Archetypal SAE: Adaptive and Stable Dictionary Learning for Concept Extraction in Large Vision Models." _Forty-second International Conference on Machine Learning_.
>
> [5] Deleuze, Gilles. "Difference and repetition." _Columbia UP_ (1994).
>
> [6] Bussmann, Bart, et al. "Learning Multi-Level Features with Matryoshka Sparse Autoencoders." _Forty-second International Conference on Machine Learning_.

---

> > ### Comment · Reviewer_929m · 2025-11-25
> > **Great rebuttal**
> >
> > I would like to thank the authors for a pretty impressive rebuttal. I sincerely appreciate the significant effort put into generating a lot of new results in such a short time frame as well as the theoretical exploration of the quadratic extension (I kinda like the appendix G, it's a great tutorial!!).
> >
> > While I still find that the specific definition is not fully motivated by a real phenomenology of the models, the empirical results speak for themselves. I believe this work represents an interesting and valuable addition to the SAE literature.
> >
> > Congratulations to the authors on a solid submission.
> > I have increased my score to reflect these improvements. Good luck for acceptance!

---

### Official Review · Reviewer_L8Fx · 2025-11-06

**Soundness:** 2
**Presentation:** 3
**Contribution:** 2
**Rating:** 2
**Confidence:** 4

**Summary:**

The current paper proposes to cluster *differences* in activations of a neural network, instead of raw activations itself, to interpret its behavior. Across a series of models and modalities, the paper shows improved probing performance, consistency of identified "features", and good steering ability.

**Strengths:**

I really like the perspective of engaging with the rich literature on concepts in philosophy and using that to motivate interpretability approaches, but I wish the paper went deeper on this narrative.

**Weaknesses:**

Reading through, I was first quite excited about the paper's idea and narrative (to look at activation differences), but, as it currently stands, I think the paper's operationalization of its core idea falls short of its promise. Main apprehensions are listed below.

- Use of clustering to define "concepts": The paper currently takes differences of activations and simply run a clustering protocol (Kmeans) to extract "concepts" from it. While I'm not necessarily a big fan of SAEs, that approach assumes a specific model of representations (linear representation hypothesis) and hence tries to isolate "features" (directions in the learned dictionary) that interfere with each other as minimally as possible, but are also substantially larger than model dimensionality. This induces, or at least we hope that it induces, a monosemantic representation (latent code) such that we can now look at a direction in isolation to interpret what it represents. My main apprehension with the paper is that there is no such affordance achieved by the proposed method. The cluster centroids are in now way more meaningful than base activations are by default. If they are, then authors should have offered examples to demonstrate how.

- Quantitative analysis: The proposed analysis does not meaningfully relate to interpretability in my opinion. Probing on activation differences (diffs) seems as likely to succeed as base activations. Even if this is not perfectly true, probing performance on base activations helps contextualize the improvement diffs help achieve. As it stands, Table 1 has numbers that are very close to each other and I'm unsure if there's any meaningful improvement in performance.

- On consistency and clustering: If I understand correctly, Fel et al. [1] follow a similar protocol as this paper to define a dictionary learning protocol akin to SAEs (called Archetyal analysis) and show that improves consistency for theoretically meaningful reasons. At the very least a discussion is warranted, but more broadly, I think the paper already addresses the challenge of consistency that this paper claims to fix too.

[1] https://arxiv.org/abs/2502.12892

Minor comment

- I think the idea of defining concepts at differences is both philosophically and operationally solid, but the intro and writing doesn't quite do justice to this. There was a good opportunity here to highlight how existing interpretability approaches assume definitions on concepts that fit into different perspective. For example, SAEs' notion of concepts is similar to prototype theory, while more dynamic approaches (like attention probes) are heading in the direction of conceptual role semantics. The Deleuzian lens offers a rich alternative to think about and I'd have appreciated this discussion, along with other frameworks' summary. It would have been even better if an argument were made in the text for why the Deleuzian perspective is better or worse than others.

**Questions:**

In table 3, I do not understand why TopK SAEs trained on activation differences do not work as well as steering by cluster centroids. Can the authors comment what's going on here?

---

> ### Author Response · Authors · 2025-11-20
>
> We thank the reviewer for their constructive feedback. We especially appreciate the recognition of our philosophical framing and the potential of analyzing _differences in activations_ as a basis for conceptual interpretability.
>
> ### Use of clustering to define "concepts"
> Indeed the space of our approach is the "activation space" since concepts are formed from linear combination of activations. However, we cluster differences in activation (i.e. linear discriminants), not raw model activations.
> The key intuition behind our approach is that directions that most frequently discriminate data samples within model's activation should be monosemantic. In practice, our experiments show that our approach outperforms sparse autoencoders in terms of probe loss (Table 1).
> We also conducted an ablation (Table 3) that actually shows empirically the difference in terms of probe loss between our approach and a clustering directly on activations.
>
> ### Quantitative analysis
> We acknowledge that the beginning of section 3.1 was unclear on how the probe loss analysis is conducted. we reformulated this description in the article. We evaluate the probe loss on the *projection* of test data onto the learned directions. Probe loss is computed for *each* attribute independently. We then extend the formulation by [1] by aggregating the results on the different attributes.
> The significance of probe loss improvements is discussed in Appendix C, with Wilcoxon signed-rank test getting a p-value of $1.7 \cdot 10^{-26}$ for the AST-Audioset experiment for example.
>
> ### Archetypal SAE address consistency
> In section 2.1 we list 4 criteria that concepts should satisfy, but we do not claim to be the first to address them. Indeed, Archetypal SAEs [2] improve stability by constraining decoder atoms to be combinations of activations, but they remain within the sparse autoencoder paradigm. However, we added the evaluation of Archetypal SAEs in Table 1 for completeness. Their global average rank is 3.2, slightly worse than TopK-SAE (2.65) and our approach (1.65)
>
> [1] Gao, Leo, et al. "Scaling and evaluating sparse autoencoders.", ICLR 2025
>
> [2] Fel, Thomas, et al. "Archetypal SAE: Adaptive and Stable Dictionary Learning for Concept Extraction in Large Vision Models.", ICML 2025

---

### Meta-Review · Area_Chair_T6BJ · 2026-01-05

**Summary:**

This paper proposes a simple unsupervised method for extracting concepts by clustering activation differences, framed as an alternative to SAEs and motivated by a “concepts as differences” perspective. Reviewers generally found the approach easy to understand and the empirical evaluation broad across models and modalities. While several reviewers raised concerns about conceptual grounding and evaluation choices, the rebuttal substantially strengthened the paper and convinced at least one initially skeptical reviewer, shifting the overall balance toward acceptance.

Overall assessment: this is a borderline but solid paper. The method is simple, empirically competitive, and now better situated relative to SAE-based approaches. While some conceptual questions remain, the strength of the rebuttal, added baselines, and one reviewer’s clear change of opinion tip the balance in favor of acceptance.

**Reviewer Concerns:**

Main concerns:
- conceptual grounding: some reviewers questioned why clustering activation differences should yield meaningful or monosemantic concepts, and felt the philosophical framing was initially more narrative than operational.
- evaluation: reliance on label-based probe loss led to questions about whether improvements reflect genuine concept quality or alignment with dataset annotations, especially when numerical gains are modest.
- baselines: concerns were raised about fairness and completeness of SAE baselines, particularly for language models, and about missing comparisons to other simple baselines.
- steering: the “lossless steering” claim was seen as overstated without stronger selectivity or downstream behavior tests.

The rebuttal is unusually strong and responsive. The authors added several important baselines (including pretrained SAEs), expanded ablations, clarified sampling and clustering details, and provided statistical significance testing. They also engaged seriously with theoretical criticisms, including extending the analysis beyond the simplest discriminant assumptions and clarifying the intended scope of the Deleuzian framing. One reviewer explicitly describes the rebuttal as impressive and increases their score, noting that the empirical results now speak more clearly for themselves. While not all conceptual concerns are fully resolved, the rebuttal meaningfully improves both rigor and positioning.

**Reviewer Scores:**

Pre-rebuttal scores were mostly borderline, with one clear reject. After the rebuttal, the distribution shifts upward.
- reviewer L8Fx: 2, possible silight increase?
- reviewer 929m: 4 → increased after rebuttal (explicitly stated)
- reviewer rHVe: 4, likely unchanged
- reviewer Jr9h: 4, likely unchanged or small increase

---

### Decision · Program_Chairs · 2026-01-26

Accept (Poster)